# Symplectic Spectrum Gaussian Processes: Learning Hamiltonians from Noisy and Sparse Data

**Yusuke Tanaka**      **Tomoharu Iwata**      **Naonori Ueda**
NTT Communication Science Laboratories
`{yusuke.tanaka.rh,tomoharu.iwata.gy,naonori.ueda.fr}@hco.ntt.co.jp`

## Abstract

Hamiltonian mechanics is a well-established theory for modeling the time evolution of systems with conserved quantities (called *Hamiltonian*), such as the total energy of the system. Recent works have parameterized the Hamiltonian by machine learning models (e.g., neural networks), allowing Hamiltonian dynamics to be obtained from state trajectories without explicit mathematical modeling. However, the performance of existing models is limited as we can observe only noisy and sparse trajectories in practice. This paper proposes a probabilistic model that can learn the dynamics of conservative or dissipative systems from noisy and sparse data. We introduce a Gaussian process that incorporates the symplectic geometric structure of Hamiltonian systems, which is used as a prior distribution for estimating Hamiltonian systems with additive dissipation. We then present its spectral representation, *Symplectic Spectrum Gaussian Processes (SSGPs)*, for which we newly derive random Fourier features with symplectic structures. This allows us to construct an efficient variational inference algorithm for training the models while simulating the dynamics via ordinary differential equation solvers. Experiments on several physical systems show that SSGP offers excellent performance in predicting dynamics that follow the energy conservation or dissipation law from noisy and sparse data.

## 1 Introduction

There is great interest in the data-driven approach for learning the dynamics of physical systems. Modeling, simulating, and forecasting dynamics from data are fundamental in engineering and physical sciences; the data-driven approach also has the potential to discover new laws in physics [16, 37]. Many real-world systems can be described by ordinary differential equations (ODEs). Classical data-driven approaches make strong assumptions about the form of the equations; thus, they are not applicable if the system's equation is unknown [8, 31, 32, 36, 41]. The pioneering work of neural ordinary differential equation (NODE) [4] has been presented as a general *black-box* model for learning *vector fields*, represented by functions, to output time derivatives of the system's state; the state of the system is continuously transformed along the vector fields (see the right part of Figure 1). The Hamiltonian neural network (HNN) [14] and variants (e.g., [6, 12, 48]) allow one to learn vector fields such that the total energy of the system (called *Hamiltonian*) is conserved. This formulation is advantageous for learning dynamics that follow the basic laws of physics (i.e., the energy conservation law). Several extended models cover Hamiltonian systems with additive dissipation [25, 47]. However, neural network based models implicitly assume that a large amount of training data with high temporal resolution is available.

This paper addresses the problem of learning Hamiltonian dynamics from noisy and sparse trajectories and predicting the dynamics from an arbitrary initial condition. We illustrate our problem in Figure 1.

36th Conference on Neural Information Processing Systems (NeurIPS 2022).

The state trajectories are assumed to be *sparse* in the following senses: 1) Limited observation trials and 2) low temporal resolution (see Figure 1). One promising approach to alleviate overfitting in sparse settings is a Gaussian process (GP) [34], which allows for model learning while considering that the data contains uncertainties. GP models have been proposed for inferring unknown dynamics from trajectories [15, 27, 28], in which GPs are used for modeling the vector fields. They utilize a GP approximation with random Fourier features (RFFs) [33] to learn the dynamics via ODE solvers. The approximation is needed for the

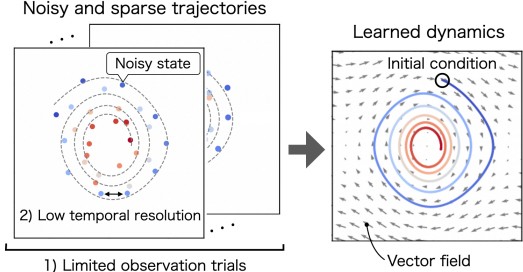

Figure 1: Problem setting. Color indicates time-evolution, starting at blue and ending at red.

following reasons. Since we can obtain only trajectory data, not derivative observations (i.e., direct observations of vector fields), we cannot use the standard GP posterior conditioned on data points; thus, we require posterior approximation. The RFF-based approximation avoids the prohibitive computational complexity of generating sample paths (i.e., realizations of the vector field) from the GP posterior (details are discussed in [44, 45]). This advantage in computation is essential for ODE-based learning. Since they do not, however, consider Hamiltonian mechanics, it is difficult to accurately capture dynamics that follow physical laws such as energy conservation and dissipation.

GP regression models for derivative observations have been proposed [38], in which the covariance function is given by matrix-valued kernels for estimating vector fields. Recently, symplectic Gaussian process regression (SympGPR) [35] has been proposed; the covariance function for learning conservative vector fields is derived by incorporating the theory of Hamiltonian mechanics into GP modeling. However, SympGPR assumes that one can obtain derivative observations, not trajectories, for training. Although it can be naively applicable to our problem by using finite differences, it is difficult to learn the dynamics accurately from noisy state observations with low temporal resolution. Also, it is not applicable to Hamiltonian systems with dissipation.

This paper proposes a GP framework that can infer Hamiltonian systems with additive dissipative terms from noisy and sparse trajectories. We first extend the GP prior for Hamiltonian dynamics proposed in [35] to handle additive dissipation. The conserved quantity of the systems (i.e., Hamiltonian) is assumed to be a single-output GP. By employing Hamilton's equations, the vector fields are derived as a multi-output GP whose covariance function incorporates the *symplectic structure* for the energy conservation law. Moreover, one can handle Hamiltonian dynamics with friction by introducing the dissipation matrix. The most significant contribution of this work is the derivation of the RFFs that encode the symplectic structure to obtain the GP approximation for dissipative Hamiltonian systems, which we call *Symplectic Spectrum Gaussian Processes (SSGPs)*. This can efficiently generate samples of the Hamiltonian vector fields from the GP posterior and construct the scalable learning algorithm based on ODE solvers. Inference in SSGP is based on variational Bayes, which allows for learning the Hamiltonian vector fields while considering the uncertainties posed by noisy and sparse data. Another benefit of SSGP is that it can be used for decomposing the dynamics into conservative and dissipative terms allowing the dynamics for unseen friction coefficients to be predicted. Such a task cannot be handled with models (e.g., NODE) that do not use prior knowledge available in physics.

The main contributions of this work are as follows:

- We introduce a GP prior for modeling Hamiltonian systems with additive dissipation.

- We propose its spectral representation (called SSGP) by deriving RFFs that incorporate the symplectic structure of Hamiltonian systems.

- We develop a variational inference procedure for SSGP that offers numerical integration by ODE solvers as a subroutine.

- Experiments on several physical systems show that SSGP can accurately predict the dynamics that follow the conservation or dissipation laws from noisy and sparse trajectories [1].

---

[1]Code is available at `https://github.com/yusuk-e/SSGP`

Table 1: Comparison of the proposed model (SSGP) with the existing models. We compare the models for four items. The first and second items represent that the models can use the physics priors, i.e., (a) energy conservation law and (b) energy dissipation law, respectively, as an inductive bias for training. The third item (c) states that the models can be learned using ODE solvers from trajectory data; the models without a checkmark require derivative observations for training. The last item (d) shows that the models can handle uncertainties present in data.

| | HNN [14] | D-HNN [9, 39] | NODE [4] | SymODEN [48] | D-SymODEN [47] | SympGPR [35] | SSGP |
|---|---|---|---|---|---|---|---|
| (a) Energy conservation law | ✓ | ✓ | | ✓ | ✓ | ✓ | ✓ |
| (b) Energy dissipation law | | ✓ | | | ✓ | | ✓ |
| (c) Learning with ODE solver | | | ✓ | ✓ | ✓ | | ✓ |
| (d) Uncertainty | | | | | | ✓ | ✓ |

## 2 Related Work

This work is positioned in the series of studies for learning *black-box* models that can estimate Hamiltonian dynamics from data. Here, black-box means that explicit mathematical modeling of differential equations is not required. We describe below prior works in this research line. Table 1 shows the comparison of the proposed model with the existing representative models.

**Neural network models.** Time-evolution of system states is generally described by differential equations. The neural ordinary differential equation (NODE) [4] and its variants [3, 19] have been proposed for learning the continuous-time evolution of the states from data. In these studies, the vector fields that determine the state evolution are parameterized by deep neural networks (DNNs) and estimated by back-propagating errors of observed trajectories via the ODE solver. The Hamiltonian neural network (HNN) [14] introduced prior knowledge of Hamiltonian mechanics as an inductive bias for training DNNs; the core concept is to parameterize the Hamiltonian (i.e., energy function) using DNNs. This formulation allows one to obtain dynamics that follow the energy conservation laws. Although the HNN assumes derivative observations for training, a learning procedure with ODE solvers from state trajectories was proposed in [6, 48]. Several models have been extended to handle Hamiltonian systems with additive dissipation [39, 47]. Most recent studies have further expanded the scope of application, including Hamiltonian systems with controllable inputs [9], stiff Hamiltonian dynamics [23], energy-conserving partial differential equation systems [25], odd-dimensional chaotic systems [7], and Poisson systems [5, 17]. Another approach, SympNets [18, 46], models the symplectic map using neural networks, which are shown to be universal approximators. However, existing DNN-based models implicitly assume a situation wherein a large amount of training data with high temporal resolution is available; they might fail to capture dynamics in the noisy and sparse settings this work focuses on.

**Gaussian process models.** Gaussian process (GP) modeling has the advantage that it makes it possible to learn models while considering uncertainties present in data [34]. Solak et al. [38] proposed GP regression models for derivative observations, in which the covariance function is given by matrix-valued kernels for inferring vector fields. Curl-free and the divergence-free kernels have been introduced for learning conservative vector fields [1, 24]; however, they lack the flexibility to be extended to cover various physical dynamics as described in the previous paragraph. Rath et al. extended the model in [38] to learn Hamiltonian vector fields, which is called symplectic Gaussian process regression (SympGPR) [35]. In SympGPR, the conserved quantity (i.e., Hamiltonian) is defined by a single-output GP; the covariance function for Hamiltonian vector fields can then be naturally derived on the basis of Hamilton's equations. However, these GP regression models assume derivative observations, not state trajectories. Although the use of finite differences allows for naively applying them to the problem of learning from trajectories, they might fail to capture dynamics, especially when the temporal resolution is lower, because each finite difference is computed from state pairs at neighboring discrete time steps. Most recently, a few models combined GPs with symplectic integrators [11, 29]; however, all the GP models [11, 29, 35] are not applicable to Hamiltonian systems with additive dissipation.

**Learning with ODE solvers.** Numerical integration methods play an essential role in learning dynamics from trajectory data. Chen et al. [4] provide a convenient tool for solving ODEs and

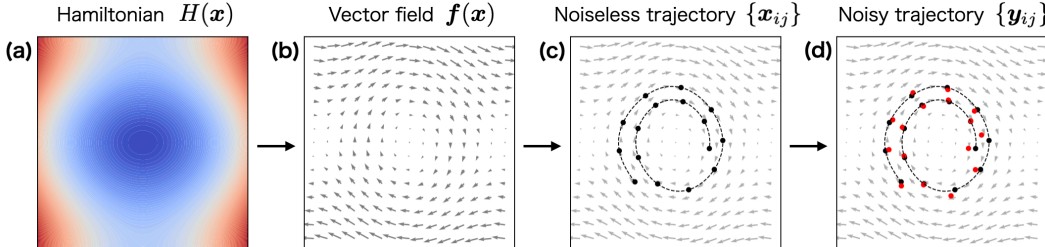

| Hamiltonian $H(\boldsymbol{x})$ | Vector field $\boldsymbol{f}(\boldsymbol{x})$ | Noiseless trajectory $\{\boldsymbol{x}_{ij}\}$ | Noisy trajectory $\{\boldsymbol{y}_{ij}\}$ |

Figure 2: Schematic diagram of SSGP: Generative processes of noisy trajectories. (a) We model the unknown Hamiltonian $H(\boldsymbol{x})$ by a single-output GP. Here, the color represents the magnitude of energy. (b) The vector field $\boldsymbol{f}(\boldsymbol{x})$ is calculated by applying the differential operator $\mathcal{L}$ to $H(\boldsymbol{x})$. (c) We sample the initial condition from the standard Gaussian distribution and solve the ODE defined by $\boldsymbol{f}(\boldsymbol{x})$ to obtain the noiseless trajectory $\{\boldsymbol{x}_{ij}\}$ depicted by black dots. (d) The noisy trajectory $\{\boldsymbol{y}_{ij}\}$ depicted by red dots is observed by adding Gaussian noise.

learning parameters via back-propagation or memory-efficient adjoint methods, which are widely used for learning Hamiltonian dynamics (e.g., [9, 12, 48]). One can also use advanced learning methods based on symplectic integrators [10, 26, 49] or discrete gradients [25], which can evaluate the loss function and its gradient while preserving energy accurately. These studies are related to ours but different in the research focus. They focus on how to estimate the parameters given the models (e.g., DNNs); whereas our proposal is a model whose learning method can be chosen as needed. It might be possible to employ the above techniques for learning our model. Different from the above studies, this work can also consider uncertainties to learn the model from noisy and sparse data.

## 3 Hamiltonian Mechanics

In this section, we briefly review Hamiltonian mechanics [13]. Let us consider a system with $N$ degrees of freedom. In the Hamiltonian formalism, the continuous-time evolution of the system is described in phase space, that is, the product space of generalized coordinates $\boldsymbol{x}^{\mathrm{q}} = (x_1^{\mathrm{q}}, \ldots, x_N^{\mathrm{q}})$ and generalized momenta $\boldsymbol{x}^{\mathrm{p}} = (x_1^{\mathrm{p}}, \ldots, x_N^{\mathrm{p}})$. Let $\boldsymbol{x} = (\boldsymbol{x}^{\mathrm{q}}, \boldsymbol{x}^{\mathrm{p}}) \in \mathbb{R}^D$ be a state of the system, where $D = 2N$. The system's evolution is determined by the Hamiltonian $H(\boldsymbol{x}) : \mathbb{R}^D \to \mathbb{R}$, which denotes the system's total energy. Traditionally, the Hamiltonian is *manually designed* to suit the system. The dynamics of a Hamiltonian system with additive dissipative terms is given by

$$\frac{d\boldsymbol{x}}{dt} = (\mathbf{S} - \mathbf{R})\nabla H(\boldsymbol{x}) =: \boldsymbol{f}(\boldsymbol{x}), \quad \text{where} \quad \mathbf{S} = \begin{pmatrix} \mathbf{O} & \mathbf{I} \\ -\mathbf{I} & \mathbf{O} \end{pmatrix}. \tag{1}$$

Here, $\nabla H(\boldsymbol{x}) : \mathbb{R}^D \to \mathbb{R}^D$ is the gradient of the Hamiltonian with respect to state $\boldsymbol{x}$, $\mathbf{S} \in \mathbb{R}^{D \times D}$ is the skew-symmetric matrix, $\mathbf{R} \in \mathbb{R}^{D \times D}$ is the positive semi-definite dissipation matrix, $\mathbf{I}$ is the identity matrix, and $\mathbf{O}$ is the zero matrix. In (1), we define the time derivatives of the state by the function $\boldsymbol{f}(\boldsymbol{x}) : \mathbb{R}^D \to \mathbb{R}^D$, which is a special kind of vector field that has a symplectic geometric structure (called Hamiltonian vector field or symplectic gradient). The dynamics on this vector field conserve the total energy when $\mathbf{R} = \mathbf{O}$. One example of the dissipation matrix is $\mathbf{R} = \mathrm{diag}(0, \ldots, 0, r_1, \ldots, r_N)$, representing a dissipative system with friction coefficient $r_n \geq 0$. Although we assume this kind of dissipation matrix in the following, this work is extendable to the general dissipation matrix, such as the state-dependent damping term [9]. Given vector field $\boldsymbol{f}(x)$ and initial condition $\boldsymbol{x}_1$ at time $t_1$, one can predict state $\boldsymbol{x}_t$ at time $t$ by integrating $\boldsymbol{f}(\boldsymbol{x})$ from $t_1$ to $t$, as follows: $\boldsymbol{x}_t = \boldsymbol{x}_1 + \int_{t_1}^t \boldsymbol{f}(\boldsymbol{x}) \, dt$.

## 4 Model

We propose SSGP (Symplectic Spectrum Gaussian Process), a probabilistic model for learning Hamiltonian systems with additive dissipation from noisy and sparse trajectories. Figure 2 shows a schematic diagram of the proposed generative processes of noisy trajectories. In the following, we first introduce a Gaussian process (GP) prior for modeling conservative and dissipative vector fields

by incorporating the theory of Hamiltonian mechanics. We then derive its spectral representation, for which we propose random Fourier features that encode symplectic structures. This spectral representation is used for constructing a scalable inference algorithm described in Section 5. Finally, we describe the generative processes of noisy state observations.

**GP priors for Hamiltonian systems with additive dissipation.** In the proposed model, the *unknown* Hamiltonian $H(\boldsymbol{x})$ is assumed to be a single-output GP with zero mean. Let $\mathcal{L} := (\mathbf{S} - \mathbf{R})\nabla$ denote a differential operator. According to (1), the vector field can be represented using $\mathcal{L}$, as follows:

$$\boldsymbol{f}(\boldsymbol{x}) = \mathcal{L}\,H(\boldsymbol{x}), \quad \text{where} \quad H(\boldsymbol{x}) \sim \mathcal{GP}(0, \gamma(\boldsymbol{x}, \boldsymbol{x}')). \tag{2}$$

Here, $\gamma(\boldsymbol{x}, \boldsymbol{x}') : \mathbb{R}^D \times \mathbb{R}^D \to \mathbb{R}$ is a covariance function. Since differentiation is a linear operator, the derivative of a GP is again a GP [38]; thus, $\boldsymbol{f}(\boldsymbol{x})$ is given by a multi-output GP,

$$\boldsymbol{f}(\boldsymbol{x}) \sim \mathcal{GP}(\mathbf{0}, \mathbf{K}(\boldsymbol{x}, \boldsymbol{x}')), \tag{3}$$

where $\mathbf{0}$ is a column vector of 0's, and $\mathbf{K}(\boldsymbol{x}, \boldsymbol{x}') : \mathbb{R}^D \times \mathbb{R}^D \to \mathbb{R}^{D \times D}$ is the matrix-valued covariance function represented by

$$\mathbf{K}(\boldsymbol{x}, \boldsymbol{x}') = \mathcal{L}\mathcal{L}^\top \gamma(\boldsymbol{x}, \boldsymbol{x}') = (\mathbf{S} - \mathbf{R})\nabla^2 (\mathbf{S} - \mathbf{R})^\top \gamma(\boldsymbol{x}, \boldsymbol{x}'). \tag{4}$$

Here, $\nabla^2$ is the Hessian operator. As described above, we can obtain the GP prior (3) with the covariance function (4) that encodes the geometric structure of Hamiltonian systems with additive dissipation. This formulation can be regarded as a generalization of the GP for non-dissipative Hamiltonian systems [35]. Though one can opt for any positive definite kernel as $\gamma(\boldsymbol{x}, \boldsymbol{x}')$, we assume that $\gamma(\boldsymbol{x}, \boldsymbol{x}')$ is shift-invariant in the approximation described in the next paragraph. One of the most widely used shift-invariant kernels is the ARD (Automatic Relevance Determination) Gaussian kernel,

$$\gamma(\boldsymbol{x}, \boldsymbol{x}') = \sigma_0^2 \exp\left(-\frac{1}{2}(\boldsymbol{x} - \boldsymbol{x}')^\top \boldsymbol{\Lambda}^{-1}(\boldsymbol{x} - \boldsymbol{x}')\right), \quad \text{where} \quad \boldsymbol{\Lambda} = \mathrm{diag}\left(\lambda_1^2, \ldots, \lambda_D^2\right). \tag{5}$$

Here, $\sigma_0^2 \in \mathbb{R}_{>0}$ and $\lambda_d^2 \in \mathbb{R}_{>0}$ are the signal variance and the length scale, respectively.

**Spectral representations.** We present the approximation of the GP prior (3), for which we newly derive random Fourier features (RFF) that encode symplectic structures of Hamiltonian systems. The RFF-based approximation is advantageous for 1) estimating the GP posterior for vector fields without derivative observations and 2) efficiently sampling the vector field from the GP posterior, which has been suggested in recent studies on learning ODEs [15, 27, 28, 44]. These advantages make it feasible to train our GP model for Hamiltonian systems with dissipation by utilizing ODE solvers, even when only trajectory data, not derivative observations, are available (see also the **On computational complexity** paragraph in Section 5 and Appendix E). We begin by modeling the Hamiltonian $H(\boldsymbol{x})$ as a GP approximation,

$$H(\boldsymbol{x}) = \sum_{m=1}^M \boldsymbol{w}_m \boldsymbol{\phi}_m(\boldsymbol{x}), \quad \text{where} \quad \boldsymbol{w}_m \sim \mathcal{N}\left(\mathbf{0}, \frac{\sigma_0^2}{M}\mathbf{I}\right). \tag{6}$$

Here, we adopt the $M$ pairs of basis functions[2], $\boldsymbol{\phi}_m(\boldsymbol{x}) = \left[\cos(2\pi \boldsymbol{s}_m^\top \boldsymbol{x}), \sin(2\pi \boldsymbol{s}_m^\top \boldsymbol{x})\right]^\top$, parameterized by a $D$-dimensional column vector $\boldsymbol{s}_m$ of spectral points. The point $\boldsymbol{s}_m$ is sampled from the kernel's spectral density, which is given by

$$p(\boldsymbol{s}) = \mathcal{N}\left(\mathbf{0}, (4\pi^2 \boldsymbol{\Lambda})^{-1}\right), \tag{7}$$

in the case in which the ARD Gaussian kernel (5) is used [22]. In (6), $\boldsymbol{w}_m \in \mathbb{R}^2$ is a row vector of weights for the $m$th pair of basis. We apply the differential operator $\mathcal{L}$ to the approximation (6); which yields the spectral representation (i.e., SSGP) of (3), represented by

$$\boldsymbol{f}(\boldsymbol{x}) = \mathcal{L}\,H(\boldsymbol{x}) =: \boldsymbol{\Psi}(\boldsymbol{x})\boldsymbol{w}^\top, \tag{8}$$

where $\boldsymbol{w} = (\boldsymbol{w}_1, \ldots, \boldsymbol{w}_M)$, and the resulting feature maps are defined by $\boldsymbol{\Psi}(\boldsymbol{x}) = (\boldsymbol{\Psi}_1(\boldsymbol{x}), \ldots, \boldsymbol{\Psi}_M(\boldsymbol{x}))$; the $m$th feature map $\boldsymbol{\Psi}_m(\boldsymbol{x}) : \mathbb{R}^D \to \mathbb{R}^{D \times 2}$ is given by

$$\boldsymbol{\Psi}_m(\boldsymbol{x}) = 2\pi(\mathbf{S} - \mathbf{R})\boldsymbol{s}_m \left[-\sin(2\pi \boldsymbol{s}_m^\top \boldsymbol{x}), \cos(2\pi \boldsymbol{s}_m^\top \boldsymbol{x})\right], \tag{9}$$

---

[2]An alternative version of basis functions [40] is known, and our proposal can easily adopt the alternative.

which we call *symplectic random Fourier features (S-RFF)*. The derivation of (8) is described in Appendix A. S-RFF (9) is the first to incorporate symplectic structures into the random features for modeling vector fields by leveraging the knowledge of Hamiltonian mechanics. This result is not trivial in that we bridge two clearly distinct research topics, random features for kernel machines (i.e., GPs) and Hamiltonian mechanics. Let us explore the connection between the exact GP (3) and its spectral approximation (8). Denoting Dirac's delta function by $p(\boldsymbol{f}|\boldsymbol{w})$, the distribution of $\boldsymbol{f}$ is given by integrating out $\boldsymbol{w}$ from (8), as follows[3]:

$$p(\boldsymbol{f}) = \int p(\boldsymbol{f}|\boldsymbol{w})p(\boldsymbol{w})d\boldsymbol{w} = \mathcal{N}\left(\boldsymbol{0}, \tilde{\mathbf{K}}(\boldsymbol{x}, \boldsymbol{x}')\right), \quad \text{where} \quad \tilde{\mathbf{K}}(\boldsymbol{x}, \boldsymbol{x}') = \frac{\sigma_0^2}{M}\boldsymbol{\Psi}(\boldsymbol{x})\boldsymbol{\Psi}(\boldsymbol{x}')^\top. \quad (10)$$

Here, $\tilde{\mathbf{K}}(\boldsymbol{x}, \boldsymbol{x}') : \mathbb{R}^D \times \mathbb{R}^D \to \mathbb{R}^{D \times D}$ is the covariance function for GP approximation. We observed that approximation quality improves with the number of spectral points (i.e., the number of basis functions) by comparing the gram matrix of $\mathbf{K}(\boldsymbol{x}, \boldsymbol{x}')$ (4) with that of $\tilde{\mathbf{K}}(\boldsymbol{x}, \boldsymbol{x}')$ in (10), where the spectral points $\boldsymbol{s}_m$ are sampled from (7). We show the gram matrices in Appendix B.

**Generative processes of noisy observations.** Suppose that we have a collection of $I$ trajectories $\{(t_{ij}, \boldsymbol{y}_{ij})|i = 1, \ldots, I; j = 1, \ldots, J_i\}$, where $J_i$ is the number of samples in the $i$th trajectory. Each sample is specified by a pair $(t_{ij}, \boldsymbol{y}_{ij})$, which represents the observation of noisy state $\boldsymbol{y}_{ij}$ at time $t_{ij}$. We treat the noiseless state $\boldsymbol{x}_{ij}$, the counterpart of $\boldsymbol{y}_{ij}$, as a latent variable. We assume that the observation model of $\boldsymbol{y}_{ij}$ is a Gaussian distribution with a variance of $\sigma^2$. Letting $\mathbf{Y} = \{\boldsymbol{y}_{ij}\}$, the marginal likelihood (i.e., evidence) is given by

$$p(\mathbf{Y}) = \int p(\boldsymbol{f})\prod_{i=1}^{I}\left[\int p(\boldsymbol{y}_{i1}|\boldsymbol{x}_{i1})p(\boldsymbol{x}_{i1})\prod_{j=2}^{J_i} p(\boldsymbol{y}_{ij}|\boldsymbol{x}_{ij})p(\boldsymbol{x}_{ij}|\boldsymbol{f}, \boldsymbol{x}_{i1})d\boldsymbol{x}_{i1}\right] d\boldsymbol{f}, \quad (11)$$

where $p(\boldsymbol{f})$ is the GP prior (10) of the vector field, and $p(\boldsymbol{x}_{i1})$ is the prior distribution[4] of the initial condition $\boldsymbol{x}_{i1}$. Given $\boldsymbol{f}$ and $\boldsymbol{x}_{i1}$, the state $\boldsymbol{x}_{ij}$ is deterministically given by solving the ODE; thus, we can write the conditional distribution $p(\boldsymbol{x}_{ij}|\boldsymbol{f}, \boldsymbol{x}_{i1})$ in (11) using Dirac's delta function, as follows:

$$p(\boldsymbol{x}_{ij}|\boldsymbol{f}, \boldsymbol{x}_{i1}) = \delta\left(\boldsymbol{x}_{ij} - \left[\boldsymbol{x}_{i1} + \int_{t_{i1}}^{t_{ij}} \boldsymbol{f}(\boldsymbol{x})\,dt\right]\right). \quad (12)$$

Note that, although we omit the observation time points $\{t_{ij}\}$ in (11), it is actually conditioned on $\{t_{ij}\}$. For simplicity, we adopt this notation hereinafter.

## 5 Inference

Although we would like to estimate the model parameters $\boldsymbol{w}$, $\boldsymbol{\Lambda}$, $\sigma_0^2$, $\sigma^2$ and $\mathbf{R}$ by maximizing the logarithm of (11), we cannot calculate the exact marginal likelihood (11) analytically as it includes the process of solving the ODEs. We then present a variational inference procedure for learning Hamiltonian systems with additive dissipation from noisy and sparse trajectories. The spectral representation of SSGP is used for efficient sampling of the conservative or dissipative vector fields from the GP posterior, allowing the learning to proceed with ODE solvers while handling uncertainties in the vector fields.

**Parameter learning.** We consider the following evidence lower bound (ELBO),

$$\log p(\mathbf{Y}) \geq \sum_{i=1}^{I}\left[\sum_{j=1}^{J_i} \mathbb{E}_{q(\boldsymbol{x}_{ij})}\left[\log p(\boldsymbol{y}_{ij}|\boldsymbol{x}_{ij})\right] - \mathrm{KL}\left[q(\boldsymbol{x}_{i1})||p(\boldsymbol{x}_{i1})\right]\right] - \mathrm{KL}\left[q(\boldsymbol{w})||p(\boldsymbol{w})\right], \quad (13)$$

where $q(\boldsymbol{x}_{ij})$ and $q(\boldsymbol{w})$ are the variational distributions of the noiseless state $\boldsymbol{x}_{ij}$ and the weights $\boldsymbol{w}$, respectively. We assume that the variational distribution of $\boldsymbol{w}$ is given by a Gaussian distribution, $q(\boldsymbol{w}) = \mathcal{N}(\boldsymbol{b}, \mathbf{C})$, where $\boldsymbol{b} \in \mathbb{R}^{2M}$ and $\mathbf{C} \in \mathbb{R}^{2M \times 2M}$ are the mean and the covariance matrix, respectively. For $j = 1$, we assume that the variational distribution of initial condition $\boldsymbol{x}_{i1}$ is given

---

[3]The derivation procedure is similar to that described in [2, Chapter 4.5.2].

[4]For simplicity, the prior is set to the standard Gaussian distribution in the implementation.

by a Gaussian distribution with the mean being the observed state $\boldsymbol{y}_{i1}$, $q(\boldsymbol{x}_{i1}) = \mathcal{N}(\boldsymbol{y}_{i1}, \mathbf{A})$, where $\mathbf{A} \in \mathbb{R}^{D \times D}$ is the covariance matrix. For $j \geq 2$, the variational distribution of $\boldsymbol{x}_{ij}$ is given by

$$q(\boldsymbol{x}_{ij}) = \iint p(\boldsymbol{x}_{ij}|\boldsymbol{f}, \boldsymbol{x}_{i1}) \left[ \int p(\boldsymbol{f}|\boldsymbol{w})q(\boldsymbol{w})d\boldsymbol{w} \right] q(\boldsymbol{x}_{i1})d\boldsymbol{x}_{i1}d\boldsymbol{f}, \tag{14}$$

where the factor in square brackets is the variational distribution of $\boldsymbol{f}$. The derivation of (13) is described in Appendix C. We approximate the expectation in (13) using Monte Carlo integration,

$$\mathbb{E}_{q(\boldsymbol{x}_{ij})}\left[\log p\left(\boldsymbol{y}_{ij}|\boldsymbol{x}_{ij}\right)\right] \approx \frac{1}{K}\sum_{k=1}^{K}\log p(\boldsymbol{y}_{ij}|\boldsymbol{x}_{ij}^{(k)}), \tag{15}$$

where the Monte Carlo samples are given by

$$\boldsymbol{x}_{i1}^{(k)} = \boldsymbol{y}_{i1} + \sqrt{\mathbf{A}}\,\boldsymbol{\epsilon}_i^{(k)}, \quad \text{where} \quad \boldsymbol{\epsilon}_i^{(k)} \sim \mathcal{N}(\mathbf{0}, \mathbf{I}), \tag{16}$$

$$\boldsymbol{x}_{i2}^{(k)}, \ldots, \boldsymbol{x}_{iJ_i}^{(k)} = \text{ODESolve}\left(\boldsymbol{x}_{i1}^{(k)}, \boldsymbol{f}^{(k)}(\boldsymbol{x}), t_{i2}, \ldots, t_{iJ_i}\right), \tag{17}$$

where we use the reparameterization trick [20] in (16), and we perform the numerical integration by the ODE solvers (e.g., the Runge-Kutta method) in (17). Here, $K$ is the number of Monte Carlo samples to obtain (15), and $\sqrt{\cdot}$ denotes a matrix square root. $\boldsymbol{f}^{(k)}(\boldsymbol{x})$ in (17) is the sample of the vector field generated from the variational posterior of $\boldsymbol{f}$,

$$\boldsymbol{f}^{(k)}(\boldsymbol{x}) = \boldsymbol{\Psi}(\boldsymbol{x})\left[\frac{1}{L}\sum_{l=1}^{L}\boldsymbol{w}^{(k,l)}\right]^{\top}, \quad \text{where} \quad \boldsymbol{w}^{(k,l)} = \boldsymbol{b} + \sqrt{\mathbf{C}}\,\boldsymbol{\epsilon}^{(k,l)}. \tag{18}$$

Here, $L$ is the number of Monte Carlo samples to obtain (18), and $\boldsymbol{\epsilon}^{(k,l)} \sim \mathcal{N}(\mathbf{0}, \mathbf{I})$. In the spectral representation, the randomness of function $\boldsymbol{f}$ is totally controlled by the distribution of weights $\boldsymbol{w}$. Accordingly, we can optimize both model parameters and variational parameters while considering uncertainties in the vector field by sampling $\boldsymbol{w}$ at each training iteration. The inference procedure is shown in Appendix D. We can easily extend our proposed approach to learn Hamiltonians from high-dimensional data (such as images) by combining an autoencoder with an SSGP, as in [14, 42]. The states $\boldsymbol{x}$ and $\boldsymbol{x}'$ in the covariance function $\mathbf{K}(\boldsymbol{x}, \boldsymbol{x}')$ (4) are modeled as the latent vectors of the autoencoder. This formulation can be regarded as an extension of the SSGP based on deep kernel learning [43].

**Prediction.** The variational posterior, $q(\boldsymbol{f}) = \int p(\boldsymbol{f}|\boldsymbol{w})q(\boldsymbol{w})d\boldsymbol{w}$, has a closed-form solution, a Gaussian distribution whose mean function and covariance function are given by $\tilde{\boldsymbol{m}}^*(\boldsymbol{x}) = \boldsymbol{\Psi}(\boldsymbol{x})\boldsymbol{b}^{\top}$ and $\tilde{\mathbf{K}}^*(\boldsymbol{x}, \boldsymbol{x}') = \boldsymbol{\Psi}(\boldsymbol{x})\mathbf{C}\boldsymbol{\Psi}^{\top}(\boldsymbol{x}')$, respectively. One can predict the dynamics from an arbitrary initial condition by numerically integrating the mean function $\tilde{\boldsymbol{m}}^*(\boldsymbol{x})$. Also, one can evaluate the predictive uncertainty for the vector field by using the covariance function $\tilde{\mathbf{K}}^*(\boldsymbol{x}, \boldsymbol{x}')$.

**On computational complexity.** The computational bottleneck is the sampling of $\boldsymbol{w}$; the cost of computing $\sqrt{\mathbf{C}}$ in (18) is $\mathcal{O}(M^3)$. The cost increases with the number of basis functions; however, the most important thing is that the bottleneck is outside of the ODE solver. The sampling of $\boldsymbol{w}$ makes it possible to efficiently evaluate function values at arbitrary point $\boldsymbol{x}$, as in (18), and to use it in the ODE solver in (17). The advantage in computational costs is discussed in Appendix E. In addition, previous studies have shown experimentally that the GP models approximated by RFF have, in many cases, high predictive performance even when setting $M$ lower than $10^3$ (e.g., [22]). Actually, our experiments show that SSGP yields an accurate prediction using a small set of RFFs. The average training time when setting $M = 250$ was 2943.0 seconds for the dataset of the pendulum with friction; the experiments were conducted on the AMD EPYC 7313 CPU (3.0GHz).

## 6 Experiments

**Data.** We evaluated the proposed model, SSGP, using two physical systems: *pendulum*, and *Duffing oscillator*. The Hamiltonian of the pendulum system is

$$H(\boldsymbol{x}) = 2mgl(1 - \cos x^{\mathrm{q}}) + \frac{l^2(x^{\mathrm{p}})^2}{2m}, \tag{19}$$

where we denote the gravitational constant by $g$, the mass constant by $m$, and the length of the pendulum by $l$. In the experiments, we set $g = 3$ and $m = l = 1$. The Hamiltonian of the Duffing oscillator system is

$$H(\boldsymbol{x}) = \frac{1}{2}(x^{\mathrm{p}})^2 + \frac{\alpha}{2}(x^{\mathrm{q}})^2 + \frac{\beta}{4}(x^{\mathrm{q}})^4, \tag{20}$$

where we set the parameters, $\alpha = \beta = 1$. For each system, the dissipation matrix was set to $\mathbf{R} = \mathbf{O}$ (energy conservation) or $\mathbf{R} = \mathrm{diag}(0, 0.05)$ (energy dissipation). We generated trajectory data by employing a numerical integrator, i.e., the Dormand–Prince method with adaptive time-stepping, implemented in torchdiffeq[5] [3, 4]. We sampled initial conditions with total energies uniformly distributed across a predefined range. The energy ranges for pendulum and Duffing oscillator were $[1.3, 2.3]$ and $[0.5, 1.5]$, respectively. To evaluate the robustness of our model against the degree of sparsity, we prepared $\{10, 15, 20, 30, 50\}$ trajectories sampled at a frequency of $\{3, 5, 10\}$ Hz for 10 seconds, and added Gaussian noise with variance $\sigma^2 = 0.1$ to each sample. We randomly split the trajectory data and used 70% for training and 30% for validation. Here, if the number of trajectories for validation was less than five, we used five trajectories for validation and the rest for training. We generated a test set of 25 trajectories independently from training and validation sets; each test trajectory was sampled at a frequency of 100 Hz for 15 seconds. The experiments were conducted five times by resampling the training and validation sets.

**Task 1: Normal prediction.** We evaluated performance by comparing the predicted state trajectories $\{\hat{\boldsymbol{x}}_{ij}\}$ with the ground truth $\{\boldsymbol{x}_{ij}^{\mathrm{true}}\}$ (i.e., the test set described above). The evaluation metric is the mean squared error (MSE), $\frac{1}{I}\sum_{i=1}^{I}(\frac{1}{J_i}\sum_{j=1}^{J_i}\parallel \hat{\boldsymbol{x}}_{ij} - \boldsymbol{x}_{ij}^{\mathrm{true}} \parallel^2)$, where $\parallel \cdot \parallel^2$ is the Euclidean norm. In the evaluation, we used the datasets from four settings: pendulum and Duffing oscillator with or without energy dissipation.

**Task 2: Predicting the dynamics for unseen friction coefficients.** One benefit of SSGP is that it can decompose the dynamics into its conservative and dissipative terms, thus predicting the dynamics for arbitrary friction coefficients. To show this, we evaluated SSGP by the following procedure: 1) we trained the model using the dataset from pendulum or Duffing oscillator with the friction coefficients, $\mathbf{R} = \mathrm{diag}(0, 0.05)$; 2) we predicted its conservative system by using the learned Hamiltonian, where we set the friction coefficients, $\mathbf{R} = \mathbf{O}$. The evaluation metric is the same as that in **Task 1**.

**Setup of the SSGP.** We trained the model using the Adam optimizer [21] with learning rate of $10^{-3}$ for $10^4$ epochs, implemented in PyTorch [30]. We performed numerical integration by the adaptive Dormand–Prince method [3, 4] with the relative and absolute tolerances of $10^{-8}$. We set the numbers of Monte Carlo samples to $K = 1$ and $L = 100$. The number $M$ of spectral points was chosen from $\{100, 250, 500\}$ based on the validation error. We used a block-diagonal approximation of $\mathbf{C}$, namely, $q(\boldsymbol{w})$ is factorized as two factors separating the weights for the sines and cosines, where the computational complexity is the same as that of the original. We fixed the friction coefficients $\mathbf{R} = \mathbf{O}$ when training the conservative systems in **Task 1**.

**Baselines.** We compared the SSGP with the existing models (see Table 1): Hamiltonian neural network (HNN) [14], dissipative HNN (D-HNN) [39], neural ordinary differential equation (NODE) [4], symplectic ODE-Net (SymODEN) [48], dissipative SymODEN (D-SymODEN) [47] and symplectic Gaussian process regression (SympGPR) [35]. Since HNN, D-HNN and SympGPR require derivative observations for training, we used the finite difference instead. Another baseline is the GP model using standard random Fourier features (RFF), which corresponds to the case that the vector field $\boldsymbol{f}(\boldsymbol{x})$ in SSGP is modeled as a multi-output GP approximated by the standard RFF (not considering Hamiltonian mechanics). In the experiments, we call it *RFF*. The details of each baseline are described in Appendix F.

**Results.** We present the experimental results that are picked up from the case where the sampling frequency was 5 Hz. Appendix G shows all the results, including those for the frequencies of 3 and 10 Hz. Figure 3 shows MSE and standard errors for SSGP and the baselines in **Task 1** and **Task 2**. As expected, in many cases, MSE decreased as the number of trajectories increased. The performance of HNN, D-HNN and SympGPR was limited because their training was based on finite differences. NODE and RFF had large errors except for the case of Figure 3 **(b)** because they cannot use the prior knowledge of Hamiltonian mechanics as the inductive bias. SymODEN and D-SymODEN were more competitive with SSGP; nevertheless, SSGP matched or bettered their predictive performance.

---

[5]https://github.com/rtqichen/torchdiffeq (MIT License)

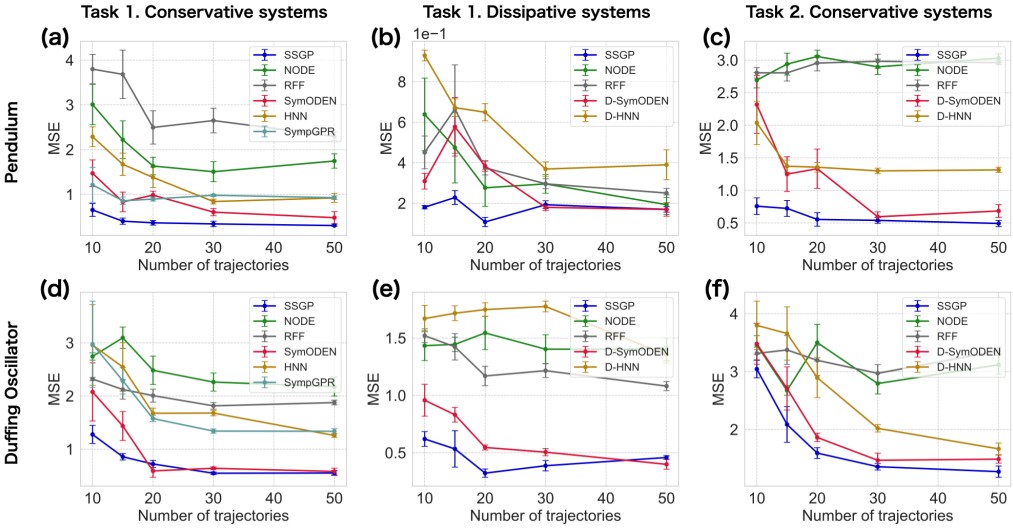

Figure 3: MSE and standard errors for the predicted trajectories when the sampling frequency was 5 Hz. The first and second columns are the results for conservative and dissipative systems, respectively, in **Task 1**. The third column holds the results predicted for conservative systems in **Task 2**.

Table 2: MSE of the friction coefficient. All values are multiplied by $10^3$.

|           | pendulum | Duffing |
|-----------|----------|---------|
| SSGP      | **0.519** | **0.506** |
| D-SymODEN | 1.010    | 0.630   |
| D-HNN     | 2.486    | 2.861   |

Moreover, SSGP improved the performance rather than all baselines, especially when the number of trajectories was small. These results show that SSGP is advantageous in sparse settings. In Figure 3 **(b)**, NODE and RFF yielded relatively low errors; however, since they cannot distinguish between the conservative and dissipative terms, their performance was worse in **Task 2**, as shown in Figure 3 **(c)**. In contrast, SSGP can estimate the conservative and dissipative terms separately by incorporating knowledge of Hamiltonian mechanics, which yields better performance in both **Task 1** and **Task 2** than all baselines. Table 2 shows MSE of the friction coefficients estimated by SSGP, D-SymODEN and D-HNN when inferring the dissipative systems. Here, MSE was averaged for all experimental settings. This result shows that SSGP can accurately estimate the friction coefficients.

In the following, we deeply explore the results when the number of observed trajectories was 20. Figure 4 shows the visualization of the trajectories predicted by SSGP. One observes that SSGP appropriately captured the dynamics of the dissipative Hamiltonian systems (the third column of Figure 4) and better discerned the conservative and dissipative terms (the fourth and fifth columns of Figure 4). Figure (5) shows the cumulative errors for the trajectories and energies for the pendulum data. Let $I^{\text{test}}$ denote the number of time steps in the test set, and $j_t$ denote a time step at time $t$. We calculated the cumulative error of trajectories at time $t$: $\sum_{j=1}^{j_t}(\frac{1}{I^{\text{test}}}\sum_{i=1}^{I^{\text{test}}} \parallel \hat{x}_{ij} - x_{ij}^{\text{true}} \parallel^2)$. The cumulative error of energies was obtained similarly by calculating the squared error of the energy, $H^{\text{true}}(\hat{x}_{ij})$, evaluated using the predicted trajectories and the true energy, $H^{\text{true}}(x_{ij}^{\text{true}})$. Here, $H^{\text{true}}(\cdot)$ is the true Hamiltonian of each system. As shown in Figure 5, SSGP achieved lower errors than the baselines for both trajectories and energies at all time points. The performance improvement of SSGP was more significant in the period $[10, 15]$, which is out of the simulation period (10 seconds) when generating the training data. These results show that SSGP can accurately predict the dynamics with energy conservation and dissipation in both short-term and long-term simulations.

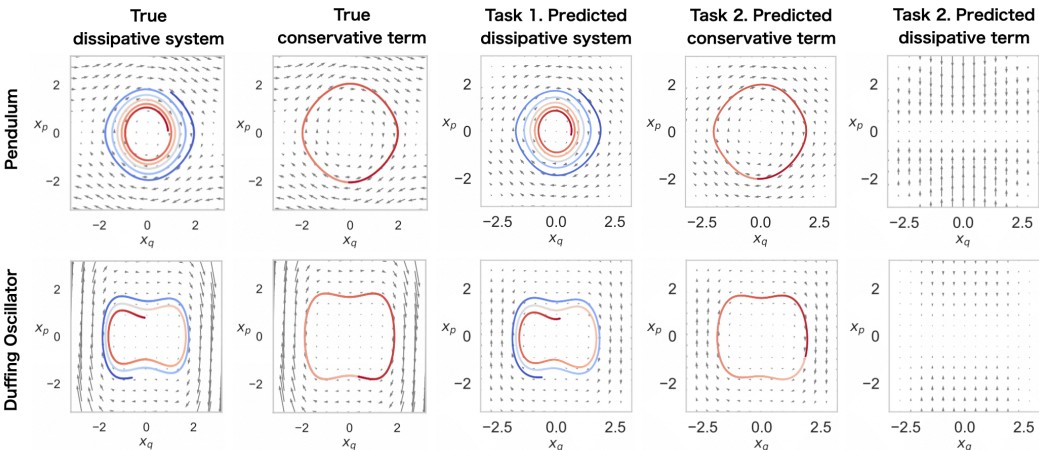

Figure 4: Prediction results of SSGP. The color indicates time-evolution, starting at blue and ending at red. The first and second columns are the true trajectories for the dissipative systems and their conservative terms, respectively. The third column is the prediction for the dissipative systems in **Task 1**. The fourth and fifth columns are the predicted conservative and dissipative terms in **Task 2**, respectively. Here, the dissipative terms are multiplied by 30 for enhanced clarity. Comparisons with other models are shown in Appendix G.

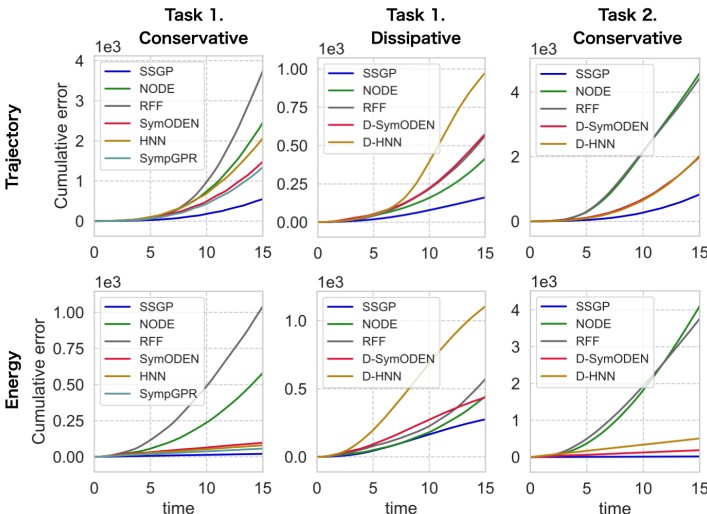

Figure 5: Cumulative error for the predicted trajectories and energies when using pendulum data. The horizontal axis is simulation time in the test phase.

## 7 Conclusion

We have proposed the Symplectic Spectrum Gaussian Process (SSGP), which allows one to predict systems whose dynamics follow energy conservation and dissipation laws from noisy and sparse data. Our result, the symplectic random Fourier feature, is a general tool and has the potential to use the design of kernel machines with prior knowledge in physics. As described in Section 5, the SSGP can easily extend to learn Hamiltonians from high-dimensional data (e.g., images). Our future work is to evaluate the effectiveness of its extension.

## Acknowledgments and Disclosure of Funding

This work was supported by JST, ACT-X Grant Number JPMJAX210D, Japan.

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
