# Supplementary Material: "Symplectic Spectrum Gaussian Processes: Learning Hamiltonians from Noisy and Sparse Data"

## A Derivation of the spectral representation

The spectral representation of the GP prior (3) for Hamiltonian systems with additive dissipation can be derived by applying the differential operator $\mathcal{L} = (\mathbf{R} - \mathbf{S})\nabla$ to the $2M$-dimensional GP approximation (6) of the Hamiltonian $H(\boldsymbol{x})$, as follows:

$$\boldsymbol{f}(\boldsymbol{x}) = \mathcal{L}\, H(\boldsymbol{x}) \tag{21}$$

$$= (\mathbf{S} - \mathbf{R})\nabla \left[ \sum_{m=1}^{M} \boldsymbol{w}_m \boldsymbol{\phi}_m(\boldsymbol{x}) \right] \tag{22}$$

$$= (\mathbf{S} - \mathbf{R}) \sum_{m=1}^{M} \boldsymbol{w}_m \nabla \left[ \cos\left(2\pi \boldsymbol{s}_m^\top \boldsymbol{x}\right), \sin\left(2\pi \boldsymbol{s}_m^\top \boldsymbol{x}\right) \right] \tag{23}$$

$$= \sum_{m=1}^{M} 2\pi (\mathbf{S} - \mathbf{R}) \boldsymbol{s}_m \left[ -\sin(2\pi \boldsymbol{s}_m^\top \boldsymbol{x}), \cos(2\pi \boldsymbol{s}_m^\top \boldsymbol{x}) \right] \boldsymbol{w}_m^\top \tag{24}$$

$$= \sum_{m=1}^{M} \boldsymbol{\Psi}_m(\boldsymbol{x}) \boldsymbol{w}_m^\top \tag{25}$$

$$= \boldsymbol{\Psi}(\boldsymbol{x}) \boldsymbol{w}^\top. \tag{26}$$

As shown in (10), the spectral representation of $\boldsymbol{f}(\boldsymbol{x})$ corresponds to the GP approximation equipped with covariance function $\tilde{\mathbf{K}}(\boldsymbol{x}, \boldsymbol{x}')$ arising from *symplectic* random Fourier features (S-RFFs) $\boldsymbol{\Psi}(\boldsymbol{x})$. The covariance function $\tilde{\mathbf{K}}(\boldsymbol{x}, \boldsymbol{x}')$ is given by

$$\tilde{\mathbf{K}}(\boldsymbol{x}, \boldsymbol{x}') = \frac{\sigma_0^2}{M} \boldsymbol{\Psi}(\boldsymbol{x}) \boldsymbol{\Psi}(\boldsymbol{x}')^\top \tag{27}$$

$$= \frac{\sigma_0^2}{M} \sum_{m=1}^{M} \boldsymbol{\Psi}_m(\boldsymbol{x}) \boldsymbol{\Psi}_m(\boldsymbol{x}')^\top \tag{28}$$

$$= \frac{(2\pi)^2 \sigma_0^2}{M} \sum_{m=1}^{M} (\mathbf{S} - \mathbf{R}) \boldsymbol{s}_m \left[ (\mathbf{S} - \mathbf{R}) \boldsymbol{s}_m \right]^\top \cos\left(2\pi \boldsymbol{s}_m^\top (\boldsymbol{x} - \boldsymbol{x}')\right). \tag{29}$$

One observes that $\tilde{\mathbf{K}}(\boldsymbol{x}, \boldsymbol{x}')$ is an extention of the covariance function arising from *standard* RFFs [22, p. 1870] to encode the geometric structure of Hamiltonian systems with additive dissipation.

## B Comparisons of gram matrices

To investigate the connection between the exact GP (3) for Hamiltonian systems with additive dissipation and its spectral approximation (8), we show the comparisons of the gram matrix of $\mathbf{K}(\boldsymbol{x}, \boldsymbol{x}')$ (4) and that of $\tilde{\mathbf{K}}(\boldsymbol{x}, \boldsymbol{x}')$ (10) in Figure 6. We observe that approximation quality improves as the number $M$ of spectral points increases.

## C Derivation of the ELBO

In this appendix, we derive the evidence lower bound (ELBO) (13). Let $\mathbf{X} = \{\boldsymbol{x}_{ij}\}$ denote the set of the noiseless states. We assume that the variational distribution is given by

$$q\left(\mathbf{X}, \boldsymbol{f}, \boldsymbol{w}\right) = p(\boldsymbol{f}|\boldsymbol{w})q(\boldsymbol{w}) \prod_{i=1}^{I} \left[ q(\boldsymbol{x}_{i1}) \prod_{j=2}^{J_i} p(\boldsymbol{x}_{ij}|\boldsymbol{f}, \boldsymbol{x}_{i1}) \right]. \tag{30}$$

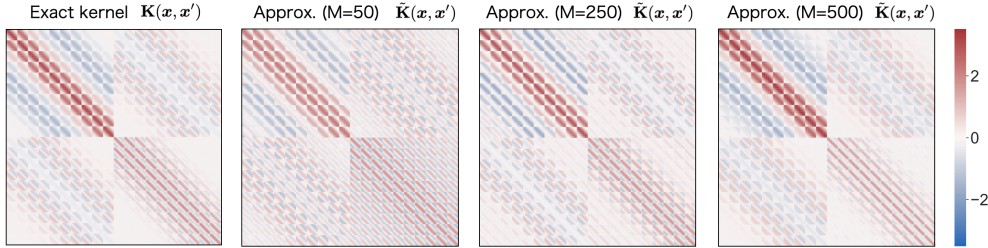

Figure 6: Gram matrices for the exact kernel and for its approximation with $M = \{50, 250, 500\}$ spectral points. The covariances are computed using $10 \times 10$ regular grid points $\{(x^{\mathrm{q}}, x^{\mathrm{p}})\}$ on a two-dimensional phase space defined by $[-1, 1]^2$. The upper left and lower right corners of each matrix are the covariances of $(x^{\mathrm{q}}, x^{\mathrm{q}'})$ and $(x^{\mathrm{p}}, x^{\mathrm{p}'})$, respectively; the remaining corners are the cross-covariances of $(x^{\mathrm{q}}, x^{\mathrm{p}})$. The parameters are set as follows: $\sigma_0^2 = 1, \boldsymbol{\Lambda} = \mathrm{diag}(0.5, 0.3), \mathbf{R} = \mathrm{diag}(0, 0.1)$.

As described in Section 5, $p(\boldsymbol{f}|\boldsymbol{w})$ is Dirac's delta function; $q(\boldsymbol{x}_{i1})$ and $q(\boldsymbol{w})$ are the variational distributions of the initial condition $\boldsymbol{x}_{i1}$ and the weights $\boldsymbol{w}$, respectively. $p(\boldsymbol{x}_{ij}|\boldsymbol{f}, \boldsymbol{x}_{i1})$ is Dirac's delta function represented by (12). The ELBO is derived from Jensen's inequality as follows:

$$\log p(\mathbf{Y})$$

$$\geq \iiint q(\mathbf{X}, \boldsymbol{f}, \boldsymbol{w}) \log \frac{p(\mathbf{Y}, \mathbf{X}, \boldsymbol{f}, \boldsymbol{w})}{q(\mathbf{X}, \boldsymbol{f}, \boldsymbol{w})} d\boldsymbol{w} d\boldsymbol{f} d\mathbf{X} \tag{31}$$

$$= \iiint p(\boldsymbol{f}|\boldsymbol{w})q(\boldsymbol{w}) \prod_{i=1}^{I} \left[ q(\boldsymbol{x}_{i1}) \prod_{j=2}^{J_i} p(\boldsymbol{x}_{ij}|\boldsymbol{f}, \boldsymbol{x}_{i1}) \right]$$

$$\times \log \frac{p(\boldsymbol{f}|\boldsymbol{w})p(\boldsymbol{w}) \prod_{i=1}^{I} \left[ p(\boldsymbol{y}_{i1}|\boldsymbol{x}_{i1})p(\boldsymbol{x}_{i1}) \prod_{j=2}^{J_i} \{p(\boldsymbol{y}_{ij}|\boldsymbol{x}_{ij})p(\boldsymbol{x}_{ij}|\boldsymbol{f}, \boldsymbol{x}_{i1})\} \right]}{p(\boldsymbol{f}|\boldsymbol{w})q(\boldsymbol{w}) \prod_{i=1}^{I} \left[ q(\boldsymbol{x}_{i1}) \prod_{j=2}^{J_i} p(\boldsymbol{x}_{ij}|\boldsymbol{f}, \boldsymbol{x}_{i1}) \right]} d\boldsymbol{w} d\boldsymbol{f} d\mathbf{X} \tag{32}$$

$$= \sum_{i=1}^{I} \left[ \int \left\{ q(\boldsymbol{x}_{i1}) \log p(\boldsymbol{y}_{i1}|\boldsymbol{x}_{i1}) + \sum_{j=2}^{J_i} \int p(\boldsymbol{x}_{ij}|\boldsymbol{f}, \boldsymbol{x}_{i1})q(\boldsymbol{f})q(\boldsymbol{x}_{i1}) \log p(\boldsymbol{y}_{ij}|\boldsymbol{x}_{ij}) d\boldsymbol{f} \right\} d\mathbf{X} \right]$$

$$- \sum_{i=1}^{I} \mathrm{KL}\left[ q(\boldsymbol{x}_{i1}) || p(\boldsymbol{x}_{i1}) \right] - \mathrm{KL}\left[ q(\boldsymbol{w}) || p(\boldsymbol{w}) \right] \tag{33}$$

$$= \sum_{i=1}^{I} \left[ \sum_{j=1}^{J_i} \mathbb{E}_{q(\boldsymbol{x}_{ij})} \left[ \log p(\boldsymbol{y}_{ij}|\boldsymbol{x}_{ij}) \right] - \mathrm{KL}\left[ q(\boldsymbol{x}_{i1}) || p(\boldsymbol{x}_{i1}) \right] \right] - \mathrm{KL}\left[ q(\boldsymbol{w}) || p(\boldsymbol{w}) \right], \tag{34}$$

where $p(\boldsymbol{f}|\boldsymbol{w})$ and $p(\boldsymbol{x}_{ij}|\boldsymbol{f}, \boldsymbol{x}_{i1})$ are cancelled in (32), and we define $q(\boldsymbol{f}) = \int p(\boldsymbol{f}|\boldsymbol{w})q(\boldsymbol{w})d\boldsymbol{w}$ in (33). For $j \geq 2$, the variational distribution $q(\boldsymbol{x}_{ij})$ in (34) is given by

$$q(\boldsymbol{x}_{ij}) = \iint q(\boldsymbol{f})p(\boldsymbol{x}_{ij}|\boldsymbol{f}, \boldsymbol{x}_{i1})q(\boldsymbol{x}_{i1})d\boldsymbol{x}_{i1}d\boldsymbol{f}. \tag{35}$$

It should be noted that, given $\boldsymbol{f}$ and $\boldsymbol{x}_{i1}$, $\boldsymbol{x}_{ij}(j \geq 2)$ is deterministically generated by the ODE.

## D  Inference procedure of SSGP

The inference procedure of SSGP is shown in Algorithm 1. Here, $\mathrm{unif}\{1, J_i - S\}$ in Algorithm 1 is a discrete uniform distribution with support $\{1, \ldots, J_i - S\}$. Although we assume that, for simplicity, the sampling frequency $G_{\mathrm{s}}$ is constant, our algorithm can be easily extended to irregularly sampled trajectories. In addition, this algorithm requires a hyperparameter, *integration time window* $\tau$, which is often used for effectively learning ODEs when the trajectory length $J_i$ is large. In the implementation, we randomly re-select a state taken to be an initial condition from the observed samples in each iteration and calculate the ELBO by generating trajectories up to $\tau$ seconds later, as in previous works (e.g., [7, 48]). In the experiments, we set the integration time window $\tau = 1$.

**Algorithm 1** Inference procedure for SSGP.

---

**Input:** Trajectory data $\{(t_{ij}, \boldsymbol{y}_{ij})|i=1\ldots,I; j=1\ldots,J_i\}$, sampling frequency $G_{\mathrm{s}}$, number of spectral points $M$, numbers of Monte Carlo samples $K, L$, integration time window $\tau$, number of epochs $E$

**Output:** Parameters $\boldsymbol{\Lambda}, \sigma_0^2, \sigma^2, \mathbf{R}, \mathbf{A}, \boldsymbol{b}, \mathbf{C}$

1: Initialize the parameters
2: $e \leftarrow 1$
3: **repeat**
4:    /* Sample mini-batches $\mathcal{D}$ */
5:    $S \leftarrow \tau/G_{\mathrm{s}}$
6:    $\mathcal{D} \leftarrow \emptyset$
7:    **for** $i = 1,\ldots I$ **do**
8:       $s_{i1} \sim \mathrm{unif}\{1, J_i - S\}$
9:       $\mathcal{D} \leftarrow \mathcal{D} \cup \{(t_{i,s_{i1}}, \boldsymbol{y}_{i,s_{i1}}), \ldots, (t_{i,s_{i1}+S}, \boldsymbol{y}_{i,s_{i1}+S})\}$
10:    **end for**
11:
12:    /* Generate trajectories */
13:    **for** $k = 1,\ldots,K$ **do**
14:       **for** $l = 1,\ldots,L$ **do**
15:          $\boldsymbol{w}^{(k,l)} \leftarrow \boldsymbol{b} + \sqrt{\mathbf{C}}\,\boldsymbol{\epsilon}^{(k,l)}$   where  $\boldsymbol{\epsilon}^{(k,l)} \sim \mathcal{N}(\mathbf{0}, \mathbf{I})$
16:       **end for**
17:       Construct the vector field $\boldsymbol{f}^{(k)}(\boldsymbol{x})$ by (18).
18:       **for** $i = 1,\ldots,I$ **do**
19:          $\boldsymbol{x}_{i,s_{i1}}^{(k)} \leftarrow \boldsymbol{y}_{i,s_{i1}} + \sqrt{\mathbf{A}}\,\boldsymbol{\epsilon}_i^{(k)}$,   where  $\boldsymbol{\epsilon}_i^{(k)} \sim \mathcal{N}(\mathbf{0}, \mathbf{I})$
20:          $\boldsymbol{x}_{i,s_{i1}+1}^{(k)}, \ldots, \boldsymbol{x}_{i,s_{i1}+S}^{(k)} \leftarrow \mathrm{ODESolve}\left(\boldsymbol{x}_{i,s_{i1}}^{(k)}, \boldsymbol{f}^{(k)}(\boldsymbol{x}), t_{i,s_{i1}+1}, \ldots, t_{i,s_{i1}+S}\right)$
21:       **end for**
22:    **end for**
23:
24:    Update the parameters by maximizing the ELBO (13) evaluated using $\mathcal{D}$.
25:    $e \leftarrow e + 1$
26: **until** $e > E$

---

# E   On computational complexity

As described in Section 5, our inference algorithm is computationally efficient because the proposed model is based on symplectic random Fourier features (S-RFFs) and the bottleneck (i.e., sampling of weights) is outside of the ODE solver. To clarify it, let us consider the case where we do not use the RFF-based approximation. Since the derivative observations are not available, we should approximate the GP posterior for vector fields; one possible way is to introduce latent variables (similar to inducing variables) for the derivative observations, as in [A1]. Unfortunately, this technique does not reduce the cost of sampling paths from the GP posterior which scales cubically with the number of test points [44, 45]. Note that sampling paths is different from sampling function values independently at each test point. Denote the number of test points by $N_{\mathrm{t}}$, which is proportional to the inverse of step size $\Delta$ in the ODE solver, as follows: $N_{\mathrm{t}} \propto 1/\Delta$, where $\Delta$ should be set to a sufficiently small value (often lower than $10^{-3}$). Moreover, the GP posterior should be updated at each numerical integration step by conditioning all data points (i.e., test points) generated by the ODE solver. The cost of this approach results in $\mathcal{O}(N_{\mathrm{t}}^4)$. On the other hand, the cost of our inference algorithm is $\mathcal{O}(M^3)$, where $M$ is the number of spectral points (often lower than $10^3$). Accordingly, our proposed model based on RFF is also beneficial in terms of computational complexity.

# F   Baselines

In this appendix, we describe baseline models for the experiments in Section 6. In all experiments, we trained the model using the Adam optimizer with a learning rate of $10^{-3}$ and a weight decay of $10^{-4}$ for $10^4$ epochs. We also used the ODE solver, the Dormand–Prince method with adaptive time-stepping, with the relative and absolute tolerances of $10^{-8}$.

**Neural network models.** HNN can utilize the prior knowledge of energy conservation as an inductive bias for training neural networks; D-HNN is its extension to handle not only energy conservation but also energy dissipation. These models require derivative observations, not trajectories, for training; thus, we used finite differences instead in the experiments. NODE can infer unknown dynamics from trajectory data by utilizing ODE solvers, which does not consider prior knowledge in physics. SymODEN is the extension of HNN to allow the model learning from trajectory data by using ODE solvers, as in NODE. D-SymODEN can also apply to the dissipative systems. For all the models, we used a fully connected neural network with 3 layers, 200 hidden units, and tanh activations.

**Gaussian process models.** SympGPR can estimate conservative vector fields from derivative observations by considering Hamiltonian mechanics; we used finite differences for training. RFF is a baseline that uses standard random Fourier features for modeling vector fields (not considering Hamiltonian mechanics); its training is based on ODE solvers. This baseline is equivalent to [15] except for using the decoupled sampling. SSGP is our proposal, which can handle conservative and dissipative systems by considering Hamiltonian mechanics and train the model with uncertainties from trajectories using ODE solvers. For all the GP models, we used the ARD Gaussian kernel.

## G  Experimental results

This appendix contains additional results for our experiments in Section 6. Figures 7(a) and 7(b) show MSEs and standard errors for SSGP and the baselines when the sampling frequency was 3 or 10 Hz. We obtained similar results to those for 5 Hz shown in Figure 3. SSGP had a more significant performance improvement when the number of trajectories was smaller and the sampling frequency was lower (see Figure 7(a)). On the other hand, the performance of SSGP was closer to those of the baselines when the trajectory data with relatively high time resolution was available (see Figure 7(b)). These results are convincing to clarify that SSGP is more advantageous in the sparse setting.

We provide the visualization comparisons of the predicted trajectories when the sampling frequency was 5 Hz. Figures 8 and 9 show the visualization results for predicting the conservative and dissipative systems, respectively, in **Task 1**. As shown in Figure 8, NODE and RFF do not capture the energy conservation law. As one can see from the results of Duffing oscillator (see Figures 8(b) and 9(b)), HNN, D-HNN, SymODEN, and D-SymODEN failed to estimate the vector fields precisely because they suffered from overfitting to noisy observations; thus, they output false trajectories. Compared with the baselines, SSGP estimated the vector fields without overfitting and accurately predicted the trajectories with energy conservation and dissipation. Figure 10 shows the visualizations of the predicted conservative and dissipative terms in **Task 2**. One observes that SSGP accurately decomposed the dynamics into conservative and dissipative terms rather than the baseline models.

## H  Discussion

**Limitations.** Finite-dimensional approximation of Gaussian processes (GPs) via a set of random Fourier features (RFFs) is helpful for efficient sampling from GP posterior. In the proposed approach, we introduce the RFFs that encode symplectic geometric structures of Hamiltonian systems to construct the efficient variational inference algorithm for learning dynamics with energy conservation or dissipation. However, the RFF-based approximation is well-known to yield undesirable extrapolatory predictions as the number of samples increases, which is called *variance starvation* [A2, A3]. This might prevent the proposed model from learning Hamiltonian dynamics effectively. It would be promising to employ the *decoupled sampling* technique [44], as in [11, 15], which might be useful to infer unknown dynamics while avoiding the variance starvation.

Although our experimental results are encouraging, they are the first step to investigating the empirical properties of the proposed model. We will apply it to various types of systems. In addition, the proposed model is easily extendable to learning Hamiltonians from high-dimensional data (such as images), as described in Section 5. We will empirically evaluate the effectiveness of its extension.

**Societal impact.** This paper presents a framework for data-driven modeling that incorporates prior knowledge of Hamiltonian mechanics to improve predictive performance in noisy and sparse settings. Generally, a data-driven approach can have a significant impact on a wide range of applications, including climate and robotics. However, since we have not evaluated our proposed framework for

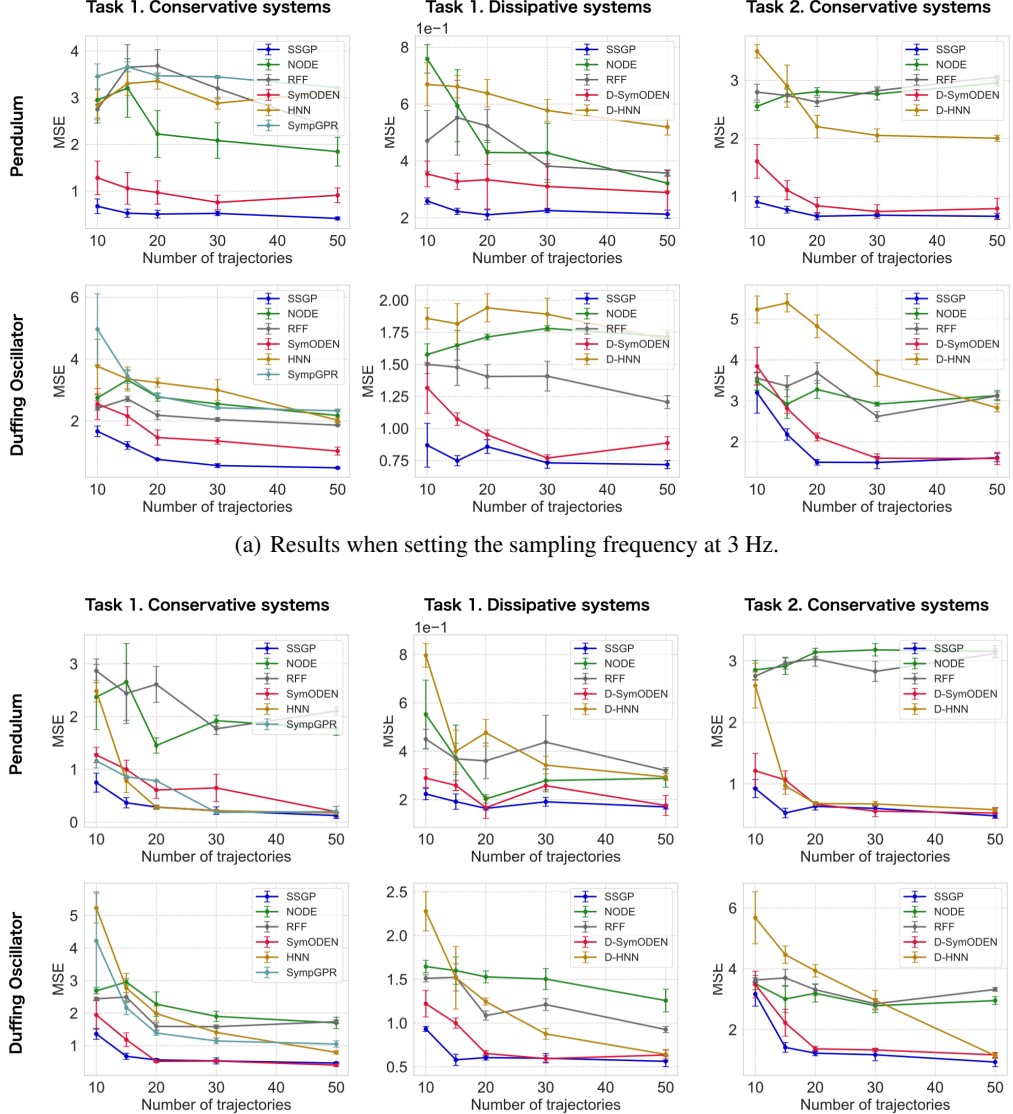

(a) Results when setting the sampling frequency at 3 Hz.

(b) Results when setting the sampling frequency at 10 Hz.

Figure 7: MSEs and standard errors for the predicted trajectories of pendulum and Duffing oscillator. We set the sampling frequency to (a) 3 Hz and (b) 10 Hz. The first and second columns are the results for the predictions of conservative and dissipative systems, respectively, in **Task 1**. The third column holds the results for predicting conservative systems in **Task 2**.

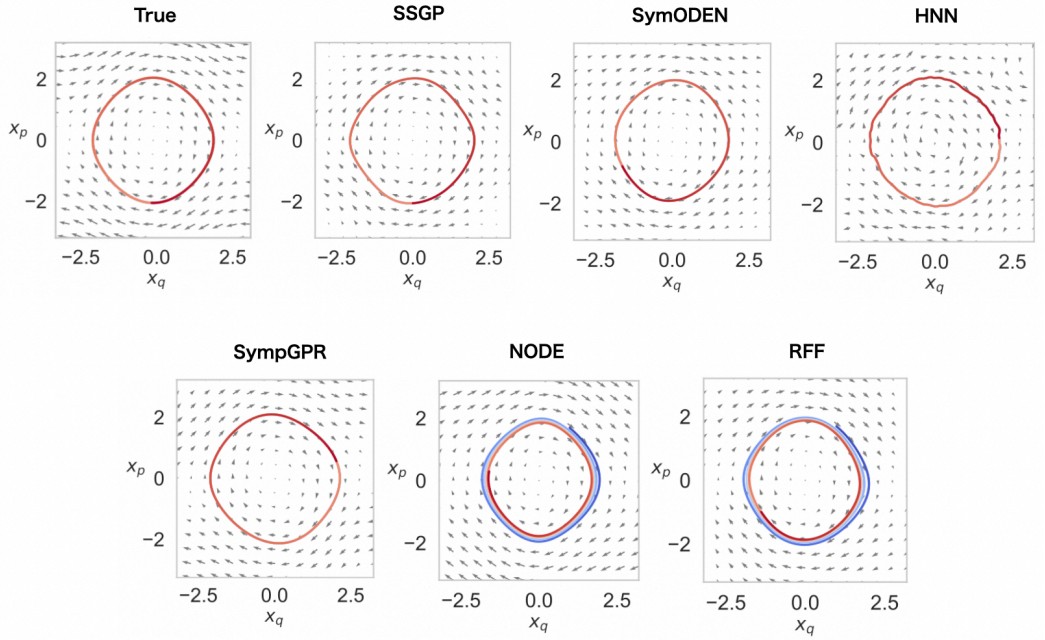

(a) Prediction results for pendulum (without dissipation)

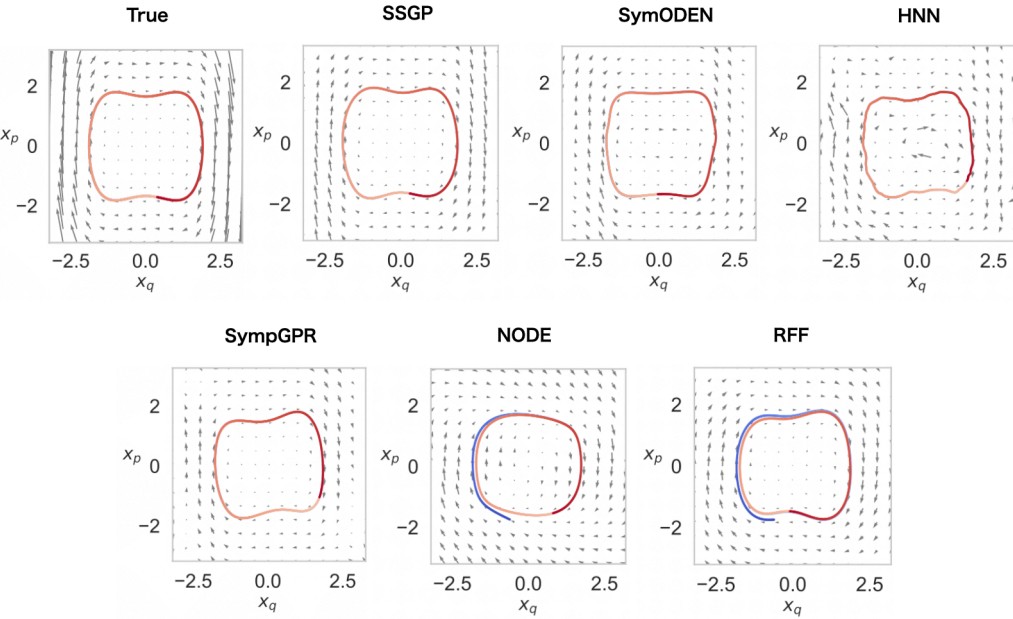

(b) Prediction results for Duffing oscillator (without dissipation)

Figure 8: Prediction results for pendulum and Duffing oscillator without dissipation in **Task 1**. The color indicates time-evolution, starting at blue and ending at red.

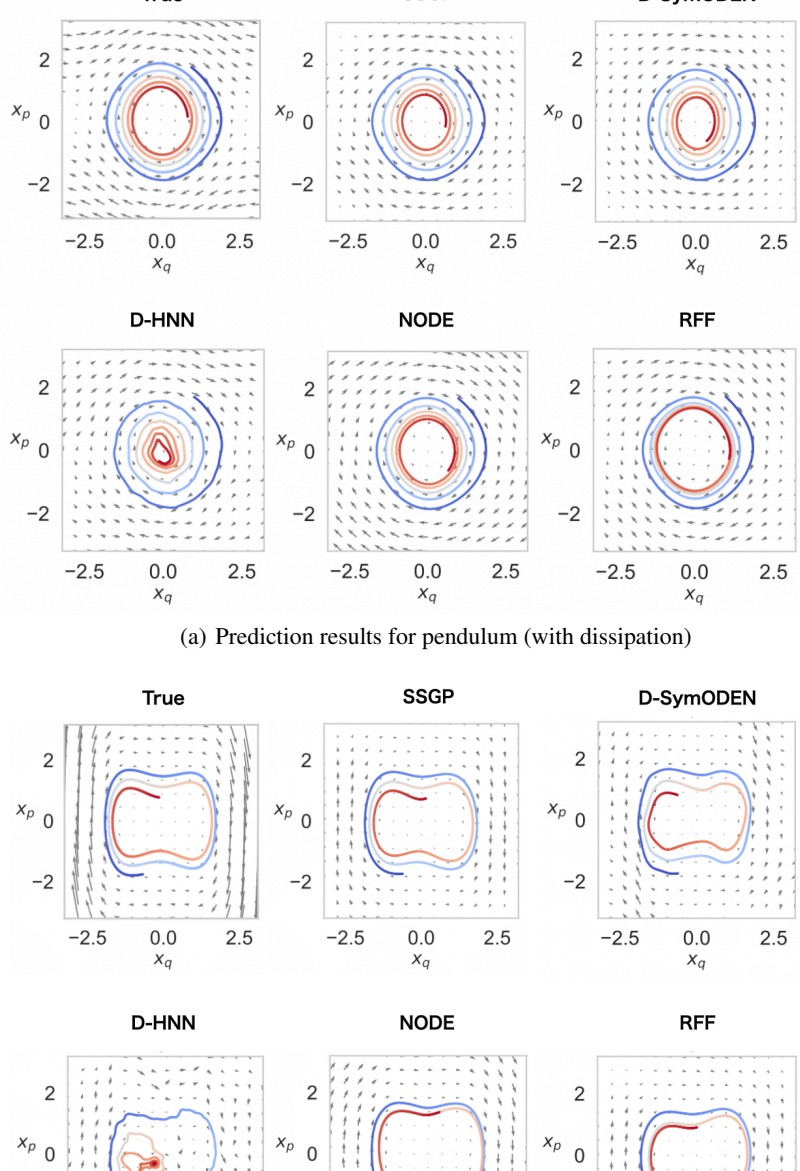

(a) Prediction results for pendulum (with dissipation)

(b) Prediction results for Duffing oscillator (with dissipation)

Figure 9: Prediction results for pendulum and Duffing oscillator with dissipation in **Task 1**. The color indicates time-evolution, starting at blue and ending at red.

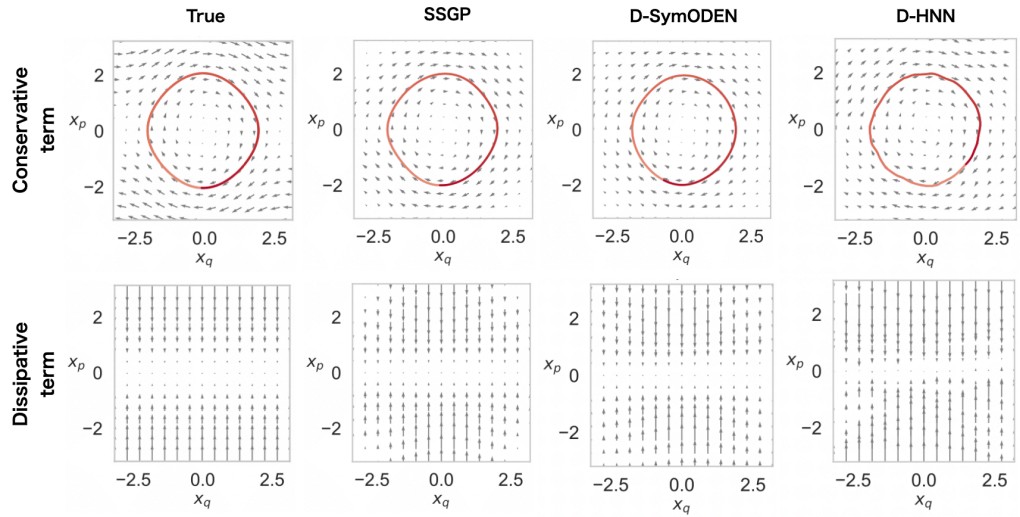

(a) Predicted conservative and dissipative terms for pendulum

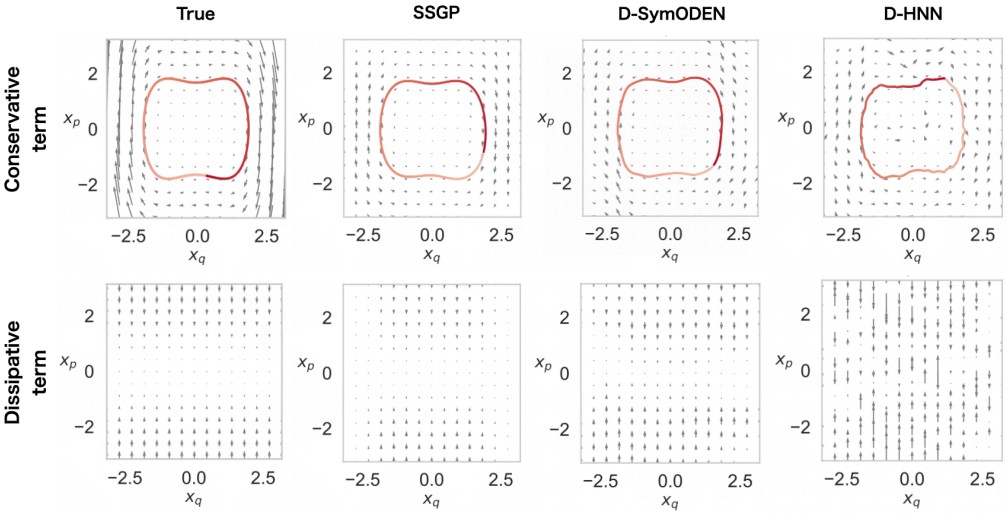

(b) Predicted conservative and dissipative terms for Duffing oscillator

Figure 10: Prediction results of the conservative and dissipative terms for pendulum and Duffing oscillator in **Task 2**. The dissipative terms are multiplied by 30 for enhanced clarity.

such real-world applications, we need to carefully assess the prediction and uncertainty estimation performance when applying it to them.