# OpenReview forum: "Symplectic Spectrum Gaussian Processes: Learning Hamiltonians from Noisy and Sparse Data"
_NeurIPS.cc/2022/Conference — NeurIPS 2022 Accept_

### Official Review · Reviewer_zQ6w · 2022-06-29

**Rating:** 6
**Confidence:** 4
**Soundness:** 3 good
**Presentation:** 3 good
**Contribution:** 4 excellent

**Summary:**

Many neural network-based methods for continuous-time physical systems have been proposed recently. These methods are based on ordinary differential equations and consistent with geometric structures. In practice, the available data is noisy and sparse. To deal with such situation, this study proposes to use Gaussian process (GP) consistent with geometric structures. Because tha data is composed only of observations (without derivatives), the GP uses the random Fourier features.


**Questions:**

(1) At line 67, the authors stated that "whose covariance function incorporates the geometric structure (also called symplectic structure) for the energy conservation or dissipation laws". The geometric structures appearing in physics are not limited to the symplectic structure, but include contact structure, Poisson structure, Dirac structure, and so on. Moreover, the symplectic structure is related to the energy conservation law, but not to the dissipation law. Please introduce and discuss the geometric structure exactly.

(2) To deal with the case without the true derivatives, this study proposes the symplectic random Fourier features. The experiments demonstrated that the features worked well for pendulum and Duffing oscillator, which exhibit periodic behaviors and are easily captured in the frequency domain. The generality for non-periodic cases such as double pendulums is unclear.

I am curious about the impacts of noise level on the accuracy. The comparison methods, namely HNN and SymODEN, ignore noise. In the noiseless case, is SymODEN enough?

*Minor comments:*

The experimental setting is a bit unclear.
(a) The difference between HNN and SymODEN is unclear. HNN learns the Hamiltonian using a single neural network, and SymODEN learns the weight matrix and the potential energy using two networks, obtaining the Hamiltonian. Is this OK?
(b) At line 298, the authors stated "Since HNN, D-HNN and SympGPR require derivative observations for training, we used the finite difference instead." Wasthe finite difference used as an approximation to the derivative?

The images are shown in the reverse order of the captions in the leftmost and second leftmost columns in Figure 4.

Due to the inexact discussion about the geometric structures, I cannot accept the present paper as it is. In this regard, I believe that a revision of the text would be sufficient. Then, I will be happy to raise the score.

**Limitations:**

As discussed in Section I (and potentially shown in Figure 10), the extrapolation performance might be limited. The authors sincerely expressed this limitation. I believe that this limitation will not diminish the value of this study.


**Strengths And Weaknesses:**

*Strengths*

To my best knowledge, this is the first study to tackle the modeling of the Hamiltonian system (with dissipation) by using GP. Like the great success of Hamiltonian neural networks (Greydanus et al., NeurIPS, 2019), this study potentially opens up a new research field.

This study tackled the observation noise, which is a crucial issue in this field, but has been ignored by most existing studies.

The impact of the sampling rate on the performance, shown in Figure 7, is interesting. The proposed method is superior especially in the case of the sparse sampling.

*Weakness*

The explanation about geometric structures is inexact. See (1) below.

The generality of the spectral representation is unclear. See (2) below.

---

> ### Author Response · Authors · 2022-08-02
> **Response to Reviewer zQ6w (2/2)**
>
> **Question 4: The experimental setting is a bit unclear. (a) The difference between HNN and SymODEN is unclear. HNN learns the Hamiltonian using a single neural network, and SymODEN learns the weight matrix and the potential energy using two networks, obtaining the Hamiltonian. Is this OK? (b) At line 298, the authors stated "Since HNN, D-HNN and SympGPR require derivative observations for training, we used the finite difference instead." Wasthe finite difference used as an approximation to the derivative?**
>
> **Response:** (a) You are right. But, the most important difference between HNN and SymODEN is whether the ODE solver can be utilized in the training process or not. The HNN is trained from the finite differences without using the ODE solver; meanwhile, the SymODEN can be trained using the ODE solver from state trajectories. I would like you to see Table 2 in Supplementary Material, which summarizes the differences between our model and the baselines.
>
> (b) You are right. These baselines assume that the time-derivative observations are available, so we used the finite differences as the approximation of the time-derivative. Please see Response to Comment 1 by Reviewer AdVL about the details of this procedure.
>
> **Question 5: The images are shown in the reverse order of the captions in the leftmost and second leftmost columns in Figure 4.**
>
> **Response:** Thank you for pointing this out. We will modify it.

---

> ### Author Response · Authors · 2022-08-02
> **Response to Reviewer zQ6w (1/2)**
>
> We would thank you for the positive evaluation of our work and your constructive feedback. Please find our response to your concerns in the following:
>
> **Question 1: At line 67, the authors stated that "whose covariance function incorporates the geometric structure (also called symplectic structure) for the energy conservation or dissipation laws". The geometric structures appearing in physics are not limited to the symplectic structure, but include contact structure, Poisson structure, Dirac structure, and so on. Moreover, the symplectic structure is related to the energy conservation law, but not to the dissipation law. Please introduce and discuss the geometric structure exactly.**
>
> **Response:** As you indicated, the description of the geometric structure was misleading. The only geometric structure considered in this study is the symplectic structure, which is appeared in the dynamics with conserved quantities. As you know, one can handle Hamiltonian dynamics with friction by introducing the dissipation matrix $R$, as in Eq. (1). To clarify it, we will modify Lines 66-68 as "By employing the theory of Hamiltonian mechanics, the vector fields are derived as a multi-output GP whose covariance function incorporates the symplectic structure for the energy conservation law. Moreover, one can handle Hamiltonian dynamics with friction by introducing the dissipation matrix.". Also, we will carefully revise the entire manuscript to avoid similar misunderstandings.
>
> **Question 2: To deal with the case without the true derivatives, this study proposes the symplectic random Fourier features. The experiments demonstrated that the features worked well for pendulum and Duffing oscillator, which exhibit periodic behaviors and are easily captured in the frequency domain. The generality for non-periodic cases such as double pendulums is unclear.**
>
> **Response:** In general, it is known that the model based on RFFs can be used for modeling non-periodic functions. The intuition of this mechanism is that the period of the function approximated by RFFs will typically be very long if the individual frequencies are not all multiples of a common base frequency. This fact has been discussed in Section 3.1 of [22]. Accordingly, the proposed model is applicable to non-periodic behaviors as well as periodic ones.
>
> We conducted the additional experiments using the double pendulum system (without friction). The MSE comparisons between the proposed model (SSGP) and the baselines (i.e., SymODEN and HNN) are shown in the following table. We prepared {10, 15, 20, 30, 50} trajectories sampled at a frequency of 5 Hz for 10 seconds, and added Gaussian noise with variance $\sigma^2=0.1$ to each sample. We took the other settings for this system from Supplementary Material of [5]. These results show that SSGP is also effective for the non-periodic double pendulum.
>
> |  | #traj.=10 | 15 | 20 | 30 | 50 |
> | :--- | ---: | ---: | ---: | ---: | ---: |
> | SSGP |  0.635  | 0.497 | 0.484 | 0.503 | 0.457 |
> | SymODEN | 1.151 | 1.092 | 1.047 | 0.996 | 0.915 |
> | HNN | 1.131 | 1.152 | 1.097 | 1.095 | 1.163 |
>
> **Question 3: I am curious about the impacts of noise level on the accuracy. The comparison methods, namely HNN and SymODEN, ignore noise. In the noiseless case, is SymODEN enough?**
>
> **Response:** We conducted the additional experiments using the pendulum (without friction) in the noiseless setting, where we set the sampling frequency to 5 Hz. We show the MSE comparisons between the SSGP and the baselines (i.e., SymODEN and HNN) in the following table. As expected, the MSEs of all the models were small rather than the result in Figure 3(a). Nevertheless, SSGP improved the predictive performance than the baselines, especially when the number of trajectories was small. This is presumably because the GP-based model makes the assumption that the vector field is smooth, which might lead to better estimation results in the noiseless but sparse data setting.
>
> | | #traj.=10 | 15 | 20 | 30 | 50 |
> | :--- | ---: | ---: | ---: | ---: | ---: |
> | SSGP | 0.135 | 0.080 | 0.073 | 0.117 | 0.060 |
> | SymODEN | 0.509 | 0.324 | 0.097 | 0.185 | 0.199 |
> | HNN | 1.862 | 1.240 | 1.125 | 0.997 | 0.993 |

---

> > ### Comment · Reviewer_zQ6w · 2022-08-05
> > **Thank you for your replies and new results.**
> >
> > The additional discussions are also reasonable. The new information resolved all my concerns.
> >
> > I reread my review again and realized that I had submitted an older version of some parts of it. At the beginning of the Strengths section, I mentioned the method as the first study of GP for Hamiltonian system. This is not true; the aim is an extension to sampled data and dissipativity. I mentioned that later in the review. Sorry for the confusion. I clarify that my score has nothing to do with this (initial) misunderstanding.
> >
> > I am willing to raise the score during the discussion with AC.

---

> > > ### Author Response · Authors · 2022-08-08
> > > **I appreciate your reply.**
> > >
> > > I am glad that your concerns have been addressed.

---

### Official Review · Reviewer_9GYm · 2022-07-08

**Rating:** 6
**Confidence:** 4
**Soundness:** 3 good
**Presentation:** 3 good
**Contribution:** 3 good

**Summary:**

This paper proposed a symplectic spectrum Gaussian process method. The method can predict systems whose dynamics follow energy conservation and dissipation laws from noisy and sparse data. The proposed method is a general tool and has the potential to use the design of kernel machines with prior knowledge in physics. Although the proposed method is solid in theory, the scenarios applied are relatively simple. In addition, some classic examples such as vortex-particle and gravitational systems should be compared with the recently proposed methods.

**Questions:**

Overall, the exposition of the paper is pretty straightforward. However, I assume it to be difficult for readers who are not familiar with Gaussian process models. More specific comments:
- The equations of the predicted systems should be given in the main text.
- Is the order of the subgraphs in Figure 4 reversed.
- The introduction of the theoretical model could be more concise.
- The idea in Figure 2 is not clear. Can you show a clear schematic?
- Why do the other methods in Figure 3 perform well in their papers?
- The selection of some parameters should be discussed in detail. For example, time step, parameter size, data set size, etc.


**Limitations:**

I would love to hear some limitations of the proposed paper about failure modes that the authors might have encountered.

**Strengths And Weaknesses:**

Strengths

This paper proposed a symplectic spectrum Gaussian process method for modeling Hamiltonian systems with additive dissipation. They derived a new spectral representation by incorporating the symplectic structure to handle energy dissipation and energy conservation systems. They also proposed a variational inference procedure that offers numerical integration of the ODE solver as a subroutine. Finally, they demonstrated some experiments on several physical systems to verify the accuracy of the proposed symplectic spectrum Gaussian process method.

Weaknesses

The demonstrated examples seem to be cherry-picked, and recently proposed methods, such as SympNet, can also accurately predict simple Hamiltonian systems.
- Adding some citations of recent papers related to symplectic neural networks would be helpful.
- The symplectic neural networks are divided into separable and non-separable. The current paper seems to discuss only separable Hamiltonian systems, which needs to be clarified.
- Limitations of the method need to be discussed.

---

> ### Author Response · Authors · 2022-08-02
> **Response to Reviewer 9GYm (2/2)**
>
> **Question 7: I would love to hear some limitations of the proposed paper about failure modes that the authors might have encountered.**
>
> **Response:** We have mentioned the limitations of the proposed models in Section I of Supplementary Material. In addition, as pointed out by Reviewer zQ6w in **Limitations**, the extrapolation performance of our model might be limited. As you can see in Figure 5, the cumulative errors of the proposed model and the baselines increased significantly in the period [10,15]. Here, the observation period was 10 seconds. Improving the accuracy of extrapolation is an interesting and challenging task, which is one of the future works. We will add this limitation in the final version.
>
> **References:**
>
> [R1] Pengzhan Jin et al., SympNets: Intrinsic structure-preserving symplectic networks for identifying Hamiltonian systems,  Neural Networks, 2020.
>
> [R2] Shiying Xiong et al., Nonseparable symplectic neural networks, ICLR, 2021.

---

> > ### Comment · Reviewer_9GYm · 2022-08-05
> > **Thank you for your reply.**
> >
> > Thank you for your very detailed reply. But I have one more concern. I don't think SympNet needs a large dataset. They already maximize the symmetry of symplectic geometry. I hope you show a complete comparison with SympNet in the next round of submissions, at least methodologically.

---

> > > ### Author Response · Authors · 2022-08-08
> > > **Thank you for your additional question.**
> > >
> > > As you indicated, SympNet incorporates the symplectic structure into the neural network architecture, which allows one to estimate Hamiltonian systems effectively from data. **One most significant difference between our model and SympNet is the modeling of observation noises: SympNet does not model the observation noises.** Since our model is based on probabilistic generative modeling (i.e., GPs), we can estimate the unknown dynamics from **noisy** observations via Bayesian inference procedures. As one can also see from Figure 2 of [R1], they did not consider the noisy trajectories as training data in their experiments. Accordingly, SympNet might degrade the performance in such **noisy and sparse settings**. We will add the above discussion in the Related Work section of the final version.

---

> > > > ### Comment · Reviewer_9GYm · 2022-08-08
> > > > **no more questions**
> > > >
> > > > Thank you! I have no more questions.

---

> > > > > ### Author Response · Authors · 2022-08-09
> > > > > **Thank you again for your valuable comments.**
> > > > >
> > > > > I am glad that your concerns have been addressed. Your comments will help us to revise our manuscript even better.

---

> ### Author Response · Authors · 2022-08-02
> **Response to Reviewer 9GYm (1/2)**
>
> We would thank you for the valuable questions and comments. Please find our response to your concerns in the following:
>
> **Comment 1: The demonstrated examples seem to be cherry-picked, and recently proposed methods, such as SympNet, can also accurately predict simple Hamiltonian systems.**
>
> **Response:** We believe that our experiments are convincing that the proposed model can improve the predictive performance, especially when the number of trajectories was small, because we conducted the evaluations in various experimental settings, varying the systems, the number of trajectories, and sampling frequency (See Figures 3 and 7). Meanwhile, we agree that it is important to discuss the cases where the proposed model does not work well. We will discuss it in Response to Question 7.
>
> As you indicated, the experiments in [R1, R2] have shown that SympNet can predict simple Hamiltonian systems. However, as described in the first paragraph of the Introduction Section, neural network-based models implicitly assume that a large amount of training data with a high temporal resolution is available. Thus, the prediction performance may degrade in the sparse data settings that this work focuses on, like the results of SymODEN (Figures 3 and 7).
>
> **Comment 2: Adding some citations of recent papers related to symplectic neural networks would be helpful.**
>
> **Response:** We will add the works of SympNet [R1, R2] to our Related Work Section. The drawback of SympNet is described in Response to Comment 1.
>
> **Comment 3: The symplectic neural networks are divided into separable and non-separable. The current paper seems to discuss only separable Hamiltonian systems, which needs to be clarified.**
>
> **Response:** Indeed, Hamiltonians can be distinguished between the separable and the non-separable cases. This means whether the system's total energy (i.e., Hamiltonian $H(x)$) can be explicitly separated into the kinetic and potential energy terms. Since our formulation is not based on this assumption, it can be used for learning dynamics that are governed by non-separable Hamiltonian as well as separable Hamiltonian.
>
> **Comment 4: Limitations of the method need to be discussed.**
>
> **Response:** Please see Response to Question 7.
>
> **Question 1: The equations of the predicted systems should be given in the main text.**
>
> **Response:** We will move Eqs. (34) and (35) in Appendix F to Section 6 of the main text.
>
> **Question 2: Is the order of the subgraphs in Figure 4 reversed.**
>
> **Response:** Thank you for pointing this out. We will modify it.
>
> **Question 3: The introduction of the theoretical model could be more concise.**
>
> **Response:** We will carefully check Sections 3, 4, and 5 and revise them to be as concise as possible. We will move Lines 194-200 to Appendix B of Supplementary Material.
>
> **Question 4: The idea in Figure 2 is not clear. Can you show a clear schematic?**
>
> **Response:** We agree that the schematic diagram of the proposed model helps to understand our formulation. So, **we have added the schematic diagram of our proposed model in Figure 11 in the revised Supplementary Material: Our idea is a novel generative model of noisy trajectories.** In the final version, we will add this diagram instead of Figure 2 in the main text and revise the manuscript as clearly as possible.
>
> **Question 5: Why do the other methods in Figure 3 perform well in their papers?**
>
> **Response:** Because they used the trajectories with a sufficient number and/or a high sampling frequency for training. For example, the SymODEN was learned using the trajectories with the sampling frequency of 20 Hz in their paper [47]. In our experiments, we used the trajectories sampled at a frequency of {3,5,10} Hz to evaluate the robustness of our proposed model against the degree of sparsity. This experimental setting is reasonable because our contribution is to present the model that can accurately predict Hamiltonian systems from noisy and sparse trajectories.
>
> **Question 6: The selection of some parameters should be discussed in detail. For example, time step, parameter size, data set size, etc.**
>
> **Response:** As described in Response to Question 5, the experimental setting was carefully designed to evaluate the predictive performance in the low data regime. We generated the training data by varying the number of trajectories (i.e., data set size) and sampling frequency (i.e., time step size). Then, we have discussed the results in detail in the Result paragraph of Section 6.
>
> The hyperparameter (i.e., parameter size) that needs to be carefully determined in the proposed model to obtain high predictive performance is the number of spectral points $M$ (i.e., the number of basis functions). We determined it automatically on the basis of the validation error (See Lines 291-292). Also, the hyperparameters of baselines (e.g., network size) have been specified in Appendix G of Supplementary Material.

---

### Official Review · Reviewer_J5NT · 2022-07-08

**Rating:** 3
**Confidence:** 2
**Soundness:** 2 fair
**Presentation:** 1 poor
**Contribution:** 2 fair

**Summary:**

Paper propose a spectral representation of Hamiltonian. Demonstrated the procedure to solve the equations of motion given a set of noisy measured trajectories (line 202).

The learning process is on the parameters used in the formalism of the special Hamiltonian (line 224).

**Questions:**

Please read the questions given in the section "Strengths and Weakness".

**Limitations:**

Discussion on limitations is given in a short paragraph towards the end of the paper. The authors propose to extend this formalism into high dimensions.

**Strengths And Weaknesses:**

Strengths:
The formalism and learning process is very different from a traditional Gaussian process approach. Hence there is some novelty in this paper.

Back information is well given and pitch at the correct level of sophistication. e.g. Hamiltonian systems, Gaussian process and literature review.


Weakness:
The main mathematical framework is given in page 5 and 6. I find that these two pages of the paper is extremely difficult to follow. This is the main weakness of this paper. I explain the reasons why these two pages are difficult to follow:
1. Eq 5. w_m is drawn from a normal distribution. In line 224, it is said that w_m is a learned parameter. Perhaps something is not explained clearly.
2. Line 195: why p(f|w) is a Dirac's delta function? How do we get Eq 9?
3. Line 208: justification needs to be given for distribution of x being a delta function Eq. 11
4. There are undefined math symbols, e.g. Eq 17, b and C
5. Does Eq 17 contradicts Eq 5?
6. The physical meanings of equations should be explained better.


It will be much better if the author rewrite the manuscript to explain more clearly and in a higher level of what their method is trying to do. After a good intuition is given to the readers, the authors may dive into details of the mathematics.

---

> ### Author Response · Authors · 2022-08-02
> **Response to Reviewer  J5NT**
>
> We would thank you for the valuable feedback on our manuscript. Based on the comments, we will revise the manuscript. **We will add the schematic diagram of the proposed model to give a good intuition. Please see Figure 11 in the revised Supplementary Material.** We will answer the questions below.
>
> **Comment 1: Eq 5. $w_m$ is drawn from a normal distribution. In line 224, it is said that $w_m$ is a learned parameter. Perhaps something is not explained clearly.**
>
> **Response:** Eq. (5) states that $w_m$ is assumed to follow a normal distribution; that is, the prior distribution of $w_m$ is a normal distribution. It should be noted that $w_m\sim \mathcal{N}(0,\frac{\sigma_0^2}{M}I)$ does not represent the realizations of $w_m$ from the prior distribution. In Section 5, given the training data, we estimate $w_m$ and the other parameters on the basis of variational Bayesian inference procedures.  As you can see from Eq. (12), the prior distribution of $w_m$ is used as a regularizer (i.e., KL divergence).
>
> **Comment 2: Line 195: why ＄p(f|w)＄ is a Dirac's delta function? How do we get Eq 9?**
>
> **Response:** The use of RFF allows us to obtain the approximation of GP represented by $f(x)=\Psi(x)w^\top$, as Eq. (7). The important thing is that the randomness of function $f(x)$ is totally controlled by the prior distribution $p(w)$. Then, given $w$, the function $f(x)$ is deterministic. In such a situation, Dirac's delta function is generally used for representing the conditional distribution of $f$, as follows: $p(f\mid w)=\delta(f-\Psi w^\top)$.
>
> The marginalization of $w_m$ in Eq. (9) is well-known to be Gaussian. One can obtain the mean and covariance of this marginal distribution by taking moments (i.e., $\mathbb{E}[f]$ and $\rm{Cov}[f]$). This result is described in Eqs. (4.148)-(4.150) of the text [2, Chapter 4.5.2], which have been mentioned in the footnote (Page 5) of our manuscript.
>
> **Comment 3: Line 208: justification needs to be given for distribution of $x$ being a delta function Eq. 11.**
>
> **Response:** The reason for using the delta function is the same as in Response to Comment 2. The sentence "the distribution of $x_{ij} (j\geq 2)$ is assumed to be Dirac's delta function" in Line 208 might be confusing. So, we will modify it as "Given $f$ and $x_{i,j-1}$, the state $x_{ij}$ is deterministically given by solving the ODE; thus, we can write the conditional distribution $p(x_{ij}\mid f, x_{i,j-1})$ using Dirac's delta function, as follows:".
>
> **Comment 4: There are undefined math symbols, e.g. Eq 17, $b$ and $C$.**
>
> **Response:** $\Psi(x)$ in Eq. (17) has been defined in Eq. (7). $w^{(k,l)}$ is the sample of the weight $w$ from the variational distribution $q(w)=\mathcal{N}(b,C)$, where $b$ and $C$ has been defined in Line 229. As the reviewer indicated, the numbers $K$ and $L$ of Monte Carlo samples were undifined in Page 6; we will specify them.
>
> **Comment 5: Does Eq 17 contradicts Eq 5?**
>
> **Response:** No. Eq. (5) represents the approximation of **GP prior** using RFFs. On the other hand, Eq. (17) represents the **GP posterior** derived by the variational inference.
>
> **Comment 6: The physical meanings of equations should be explained better.**
>
> **Response:** Our aim is to infer the unknown Hamiltonian (i.e., energy function) $H(x)$ from noisy and sparse trajectories and predict the dynamics from an arbitrary initial condition. To achieve this, we propose a novel probabilistic generative model based on GP and its Bayesian inference procedure. **To give a good intuition of our model, we have added the schematic diagram of the proposed model in Figure 11 of the revised Supplementary Material.** This diagram is the generative process of noisy trajectories (not inference procedure). Our model assumes that the observed data are generated from this generative process; then, the model parameters are estimated by the variational Bayesian method, as described in Section 5. In the final version, we will add this diagram instead of Figure 2 in the main text and revise the manuscript as clearly as possible.

---

### Official Review · Reviewer_AdVL · 2022-07-11

**Rating:** 6
**Confidence:** 4
**Soundness:** 3 good
**Presentation:** 3 good
**Contribution:** 3 good

**Summary:**

The manuscript proposes a way to learn Hamiltonian systems with additive dissipation from (scattered) observations.
The learning is framed as a (Bayesian) inference of a Hamiltonian endowed with Gaussian Process prior with the rest of the generative model relating trajectories to the hamiltonian via dissipative dynamical equations, and noisy observations on top of these trajectories.

The prior covariance matrix on trajectories is approximated using RFF for an ARD base kernel and propagating the approximation via the linear operator specifying the dynamics.

Variational inference under this (approximate) prior model is used, which conveniently allows to use the reparametrization trick to evaluate the ELBO.

The resulting method is evaluated on a number of experiments and compared to pre-existing methods.


**Questions:**

I don't understand the following sentence:
"We used a block diagonal approximation of C so that each pair of basis functions shared the same covariance".
Is this mean field $q(C)=\prod_i^M q(C_i)$, or $q(C) = \tilde{q}(C)^K$. In both case, this has strong consequence on the inference.
These choices should be discussed.


**Strengths And Weaknesses:**

-- Strengths

The method is a useful extension of the SymGPR method to dissipative systems.

The combination of the prior on hamiltonian, dissipative dynamics and RFF appears original to me.

The combination of many different approximations (RFF, VI, stochastic evaluation) leads to a practical algorithm for the problem at hand.

The possibility to make predictions without or with different dissipation is expected but neat.

The results are impressive especially in the low data regime, where the method clearly outperforms alternatives in its prediction accuracy.

-- Weaknesses

I report here a few points that made the manuscript a bit difficult to read.
It is mostly about the motivation rather than the technical content.

1) Previous work, especially the symGPR is not introduced which makes it difficult to understand the novelty here.
What did they do exactly? did they also use RFF to approximate the covariance or do the calculation closed form?

2) the many approximations introduced are not necessarily well motivated. Why do you do the RFF? Is it necessary?
 convenient for VI? for scalability? I have my guess but this needs to be more explicitely stated in the paper.

3) The consequence of the approximations introduced are not discussed.
Does the RFF preserves the symplectic structure? What is lost by approximating the prior does it bias the inference? if so how?
The same applies to using VI.


Because of these perceived weaknesses,
I set my score to weak accept. I m willing to change my evaluation if my concerns are adequately addressed

---

> ### Author Response · Authors · 2022-08-02
> **Response to Reviewer AdVL (2/2)**
>
> **Question 1: I don't understand the following sentence: "We used a block diagonal approximation of $C$ so that each pair of basis functions shared the same covariance". Is this mean field $q(C)=\prod_i q(C_i)$, or $q(C)=\hat{q}(C)^K$. In both case, this has strong consequence on the inference. These choices should be discussed.**
>
> **Response:** Assume that the set of basis functions is represented as follows: $[\cos(2\pi s_1^\top x),\ldots,\cos(2\pi s_M^\top x),\sin(2\pi s_1^\top x),\ldots,\sin(2\pi s_M^\top x)]$. Then, the variational distribution of the weights $w\in \mathbb{R}^{2M}$ is given by $q(w)=\mathcal{N}(b,C)$. To save the computational time, we used the block-diagonal approximation of $C$ as follows: $C=[[C_1,0][0,C_2]]$, where $C_1, C_2 \in \mathbb{R}^{M\times M}$. In some experiments, the prediction accuracy was almost the same as with full-matrix $C$. We will clarify this in the final version.

---

> > ### Comment · Reviewer_AdVL · 2022-08-08
> > **mean field**
> >
> > thanks for the clarification.
> > A simple way to say could be that q factorizes as two factors separating the weights for the sines and cosines.

---

> > > ### Author Response · Authors · 2022-08-09
> > > **Thank you for your reply.**
> > >
> > > You are correct. We will clarify it as you commented.

---

> ### Author Response · Authors · 2022-08-02
> **Response to Reviewer AdVL (1/2)**
>
> We would thank you for the positive evaluation of our work and your constructive feedback. Following the reviewer's comments, **we will clarify this work's novelty and motivation.** Please find our response to your concerns in the following:
>
> **Comment 1: Previous work, especially the symGPR is not introduced which makes it difficult to understand the novelty here. What did they do exactly? did they also use RFF to approximate the covariance or do the calculation closed form?**
>
> **Response:** The SympGPR assumes that derivative observations are available, where each derivative observation is a pair $(x,\dot{x})$ containing a state $x$ and its time-derivative $\dot{x}=\frac{dx}{dt}$. Then, they model the conditional probability $p(\dot{x}\mid x)$ using GPR with covariance function inspired by Hamiltonian mechanics. The training is based on the exact marginal likelihood of $\dot{x}$; one can predict the time-derivative at any state by calculating the predictive distribution and can simulate the dynamics. Notice that they did not use ODE solvers for the training phase (ODE solvers are used **only** for simulation in the test phase). Also, they did not introduce the RFF approximation and did not consider the energy dissipation.
>
> In practice, it is difficult to observe the time-derivatives directly; we often obtain state trajectories $\{(t, x)\}$ instead. Although the SympGPR is applicable by approximating the time-derivatives $\dot{x}$ with finite differences $\frac{\Delta x}{\Delta t}$, it is problematic, especially when the temporal resolution is lower (See Lines 116-119).
>
> We aim to present the algorithm for training the GP models for Hamiltonian systems (with dissipation) from state trajectories by employing ODE solvers.  Our novelties compared with the SympGPR are:
> - A GP prior for modeling systems with energy dissipation as well as energy conservation.
> - Its spectral representation by deriving RFFs that incorporate the symplectic structure.
> - A variational inference (VI) procedure with a numerical integration by ODE solvers as a subroutine.
>
> **Comment 2: the many approximations introduced are not necessarily well motivated. Why do you do the RFF? Is it necessary? convenient for VI? for scalability? I have my guess but this needs to be more explicitely stated in the paper.**
>
> **Response:** The approximations (RFF and VI) are necessary to utilize ODE solvers in the training procedures of the GP models. One of the most important reasons to adopt the RFF is scalability. In our training process, we should perform numerical integration via ODE solvers, as in Eq. (16). If we do not use the RFF approximation but the exact GP, the computational costs become prohibitive (the fourth power of the number of points evaluated by the ODE solver). We have elaborated on it in Appendix E of Supplementary Material.
>
> Also, we adopted VI because we cannot calculate the exact marginal likelihood (Eq. (10)) analytically as it includes the process of solving the ODEs. We have mentioned it in Lines 225-226. We will clarify it more in the final version.
>
> **Comment 3: The consequence of the approximations introduced are not discussed. Does the RFF preserves the symplectic structure? What is lost by approximating the prior does it bias the inference? if so how? The same applies to using VI.**
>
> **Response:** The vector field $f(x)$ approximated by RFFs always preserves the symplectic structure even if the number of basis functions is finite. This is because $f(x)$ is defined by Hamilton's equation $f(x)=\mathcal{L}H(x)$ as in Eq. (7). Meanwhile, in the case where the number of RFFs is small, the expressive power of $H(x)$ (Eq. (5))  approximating the Hamiltonian might decrease compared with the exact GP in Eq. (2).
>
> Also, VI does not prevent the symplectic structure from being satisfied. In VI, the evidence (Eq. (10)) is approximated by introducing the variational distributions $q(w)$ and $q(x_{ij})$, as Eq. (12). It may be necessary to discuss how tight the evidence lower bound (Eq. (12)) is. For example, the choice of variational distributions $q(w)$ and $q(x_{i1})$ is optional. This work's contribution is to present the VI framework for learning the GP model inspired by Hamiltonian mechanics. A more practical design of VI is one of the future works.

---

> > ### Comment · Reviewer_AdVL · 2022-08-08
> > **Thanks for your comments**
> >
> > Thanks for your comments.
> > I understand past work and motivation and they are quite clear in your response.
> > **Final suggestion**: Your manuscript would gain in clarity if you followed this presentation for the past work symgpr (maybe add the name when you cite, like (symgpr [10] )) Before you introduce VI, you could explain what would be the ideal thing to do: marginal likelihood, and why it is not tractable.
> > Then introduce VI, RFF explaining what are the gains (scalability) and what it allows: reparameterization trick + ODE solver conditioned on samples for f.
> >
> > I have no further questions

---

> > > ### Author Response · Authors · 2022-08-09
> > > **Thank you again for your valuable comments.**
> > >
> > > I appreciate your reply. I am glad that your concerns have been addressed. Your comments will help us to revise our manuscript even better. I will clarify the past work SympGPR and polish the presentation of the technical part so that the motivation for introducing the approximations is clear.

---

### Meta-Review · Area_Chair_5zTa · 2022-08-30

**Recommendation:** Accept
**Confidence:** Certain

**Metareview:**

Learning from continuous-time physical systems when input data is noisy & sparse, and without access to time derivatives, is a hard problem. The authors propose a novel algorithm using Gaussian Processes, guided by physical knowledge. Reviewers agreed that the work was original. One reviewer raised concerns about the readability of the paper. The authors' responses will likely address most of those concerns. Other reviewers also suggested a number of improvements, which the authors took on board and will easily implement. Despite relatively simple experiment scenarios, this new algorithm demonstrated some advantages in the low data regime, where it improves on previous known algorithms.

**Award:**

No

---

### Decision · Program_Chairs · 2022-09-14

Accept